# Rethinking Human Evaluation Protocol for Text-to-Video Models: Enhancing Reliability, Reproducibility, and Practicality

**Tianle Zhang**[1,2], **Langtian Ma**[3], **Yuchen Yan**[4], **Yuchen Zhang**[2], **Kai Wang**[2], **Yue Yang**[1],
**Ziyao Guo**[1], **Wenqi Shao**[1], **Yang You**[2], **Yu Qiao**[1], **Ping Luo**[5], **Kaipeng Zhang**[1]*

[1]Shanghai AI Laboratory    [2]National University of Singapore    [3]University of Wisconsin-Madison
[4]University of California San Diego    [5]The University of Hong Kong

## Abstract

Recent text-to-video (T2V) technology advancements, as demonstrated by models such as Gen2, Pika, and Sora, have significantly broadened its applicability and popularity. Despite these strides, evaluating these models poses substantial challenges. Primarily, due to the limitations inherent in automatic metrics, manual evaluation is often considered a superior method for assessing T2V generation. However, existing manual evaluation protocols face reproducibility, reliability, and practicality issues. To address these challenges, this paper introduces the Text-to-Video Human Evaluation (T2VHE) protocol, a comprehensive and standardized protocol for T2V models. The T2VHE protocol includes well-defined metrics, thorough annotator training, and an effective dynamic evaluation module. Experimental results demonstrate that this protocol not only ensures high-quality annotations but can also reduce evaluation costs by nearly 50%. We will open-source the entire setup of the T2VHE protocol, including the complete protocol workflow, the dynamic evaluation component details, and the annotation interface code[2]. This will help communities establish more sophisticated human assessment protocols.

## 1 Introduction

Text-to-video (T2V) technology has made significant advancements in the last two years and garnered increasing attention from the general community. T2V products such as Gen2 [20] and Pika [18] have attracted many users. More recently, Sora [65], a powerful T2V model from OpenAI, further heightened public anticipation for the T2V technology. Predictably, the evaluation of the T2V generation will also become increasingly important, which can guide the development of T2V and assist the public in selecting appropriate models [40, 54]. This paper does a comprehensive paper survey (see Appendix F for details) and explores a human evaluation protocol for T2V generation.

Automatic and human evaluation are the two main kinds of evaluation for video generation. In recent years, nearly half of video generation papers conduct only automatic evaluation, such as Inception Score (IS) [76], Frechet Inception Distance (FID) [34], Frechet Video Distance (FVD) [84], CLIP Similarity [70] and Video Quality Assessment (VQA) [82, 49]. However, these metrics meet various challenges, such as relying on reference videos for calculation, overlooking temporal motion changes, and, more importantly, not aligning well with human perception [67, 54]. Undoubtedly, automatic evaluation is a promising research direction, but human evaluation is more convincing so far.

---

*Coressponding author

[2]The code is available at https://github.com/ztlmememe/T2VHE.

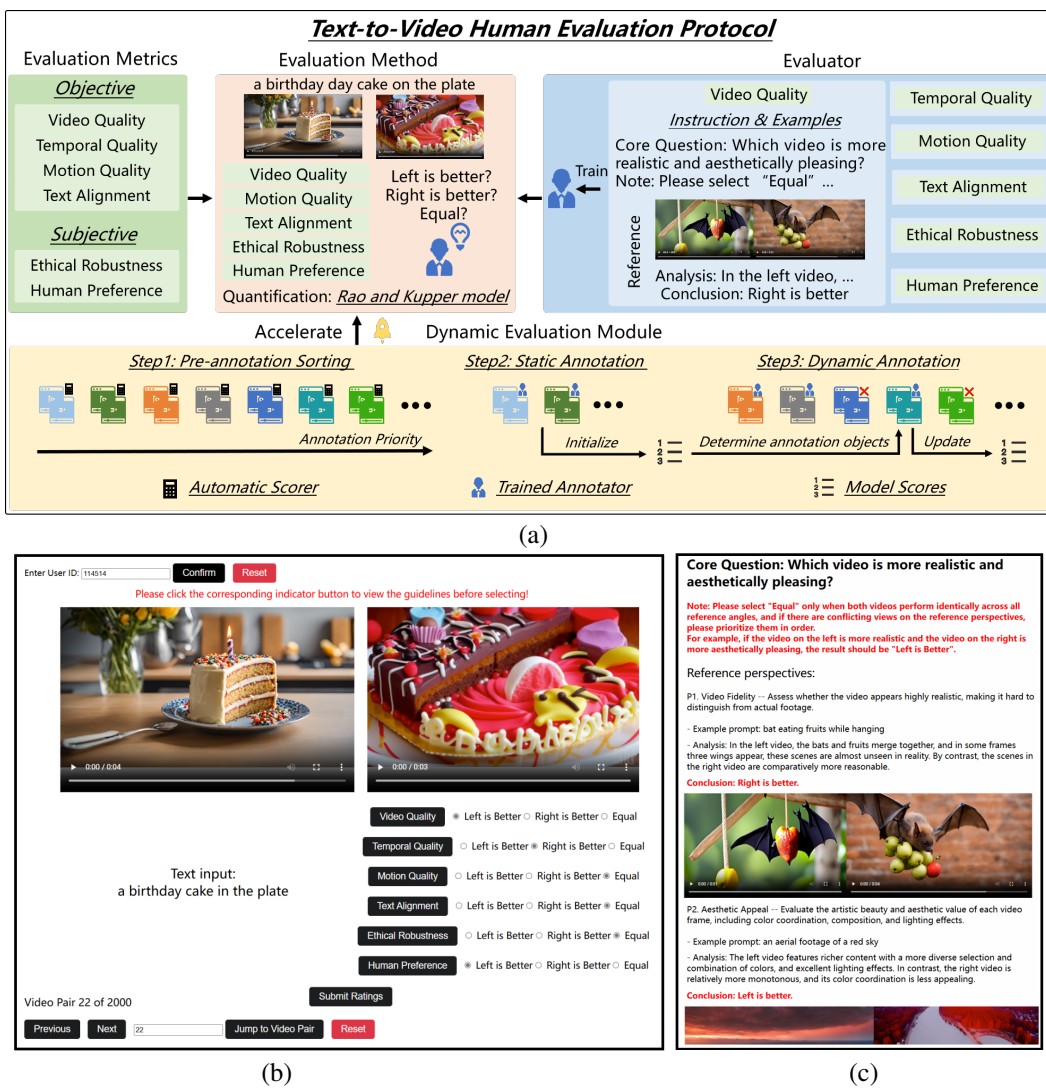

Figure 1: (a) An illustration of our human evaluation protocol. (b) The annotation interface, wherein annotators choose the superior video based on provided evaluation metrics. (c) Instruction and examples to guide used to the "Video Quality" evaluation.

Existing human evaluation for video generation also meets reproducibility, reliability, and practicability challenges. Our survey reveals that few papers employ consistent human evaluation protocols, evidenced by significant disparities in assessment metrics, evaluation methods, and annotator sources across studies. For example, some papers [100, 53] use Likert scales while others prefer comparative approach [40, 26]. Moreover, many papers lack detailed information on evaluation protocols, impeding further analysis and reproducibility. Additionally, most papers rely on annotators recruited directly by the authors, i.e., laboratory-recruited annotators (LRAs). These articles often lack quality checks, which can introduce bias and affect reliability [8]. Furthermore, there is notable variation in the number of annotations used across papers, ranging from a few dozen to tens of thousands, posing a practical challenge in achieving a balance between credible results and resource constraints.

This paper introduces Text-to-Video Human Evaluation (T2VHE), a standardized human evaluation protocol for T2V generation. T2VHE consists of well-designed evaluation metrics, annotator training encompassing detailed instructions and illustrative examples, and a user-friendly interface, see Figure 1 for illustrations. Additionally, it introduces a dynamic evaluation module to reduce the annotation cost. We further introduce the details of our protocol as follows:

**Evaluation metrics.** Many previous research focuses on video quality and text alignment but neglects the critical aspects of temporal and motion quality inherent to videos [103, 74], as well as ethical considerations [2]. T2VHE employs four objective evaluation metrics: video quality, temporal quality, motion quality, and text alignment, and two subjective metrics: ethical robustness and human preference. For different types of metrics, annotators were asked to rely more on the definitions of reference perspectives and their preferences to make judgments, respectively.

**Evaluation method.** Due to the complexities and potential noise inherent in absolute scoring [67], T2VHE employs the comparison-based method, which is relatively annotator-friendly [8]. Critiquing the reliance of traditional protocols on win rates, which may introduce biases and offer limited insights into model performance [33, 22], T2VHE adopts the Rao and Kupper model [72] to quantify annotations results. This approach enables more efficient management of pairwise comparison results, yielding improved model rankings and score estimations.

**Evaluators.** While crowdsourcing platforms are often considered to gather high-quality annotations [8, 15], our survey reveals that researchers primarily rely on LRAs. However, concerns about LRAs' reliability due to inadequate training and quality checks have been raised, potentially biasing evaluation outcomes. To address this, T2VHE proposes comprehensive annotator training, including detailed guidelines and examples to improve understanding of evaluation metrics. Experimental validation shows that properly trained LRAs can achieve agreement with crowdsourced annotators, affirming the effectiveness of our training approach in ensuring high-quality annotations.

**Dynamic evaluation module.** T2VHE incorporates a dynamic evaluation component to enhance protocol efficiency. This component optimizes utility through two key functionalities: firstly, pre-annotation sorting of videos using automatic scoring results, prioritizing the annotation of video pairs considered more deserving of manual evaluation during the static annotation phase; secondly, the dynamic annotation phase, which determines whether to annotate pending video pairs based on differences in model scores. Experimental results indicate that this module reduces costs by approximately 50%, while concurrently ensuring the validity of the annotation results.

Our study reveals several findings: (1) Post-training LRAs and annotators on crowdsourcing platforms achieve consensus. The low quality of the pre-training LRAs' annotations mainly stems from annotators' biased interpretations of evaluation metric definitions. Thus, furnishing detailed guidelines and example training can substantially improve annotation quality, yielding results comparable to those obtained by professional annotators. (2) Comparison-based evaluation exhibits significant potential for optimization. It manifests an $O(N^2)$ growth trend in the number of annotations as the number of models compared increases. Our dynamic evaluation module reveals that judiciously selecting samples for annotation can produce model ranking outcomes consistent with those derived from fully annotated data, even when annotating only approximately half of them. (3) A substantial disparity persists between open-source and closed-source models. While open-source models perform well in some evaluation metrics, videos generated by closed-source models generally exhibit superior quality and tend to garner greater popularity among annotators.

The main contributions of this paper are as follows:

1. We introduce a standardized human evaluation protocol for T2V models, comprising a meticulously crafted array of evaluation metrics alongside accompanying annotator training resources. Moreover, it lets us acquire high-quality annotations utilizingLRAs.

2. Our dynamic evaluation component reduces annotation costs to approximately half of the original expenditure while maintaining annotation quality.

3. We comprehensively evaluate the latest T2V models and commit to open-sourcing the entire evaluation process and code, empowering the community to assess new models with fresh data based on existing reviews. Our protocol is also easily extended.

# 2    Related work

**Text-to-video generative models.** Creating realistic and novel videos has been a compelling research area for many years [87, 71, 111]. Various generative models have been investigated in prior studies, including GANs [87, 75, 83, 81, 77], autoregressive models [80, 105, 61, 23, 37], and implicit neural representations [79, 113]. T2V generation focuses on producing videos from textual descriptions.

Recently, the success of diffusion models in image synthesis has spurred studies to adapt these models for conditional and unconditional video synthesis [86, 31, 123, 101, 5, 45, 107].

**Evaluation metrics of video generative models.** Evaluation metrics for video generative models can be broadly classified into automated evaluation metrics and benchmark methods. IS assesses image diversity and clarity through a pre-trained network [76], CLIP Similarity assesses alignment between textual descriptions and generated images [70], FID and FVD compare feature representations from real and generated images [34, 84]. Despite their usefulness, these metrics have limitations, including bias, sensitivity to surface similarity, and inconsistency with human perception [67, 54]. Additionally, VBench [40], EvalCrafter [53], and FETV [54], among other benchmarks, provide comprehensive evaluations but may lack the diversity to cover all real-world scenarios, and these automated scorers typically require alignment training based on human evaluation results. Given the limitations of these automated metrics and benchmarks, high-quality manual assessments remain critical.

## 3   Human evaluation in video generation

We survey 89 papers on video generation models since 2016[3], and after reviewing how they use and report human assessments, we have the following findings:

**Automatic vs. Human evaluation.** Among the 89 papers reviewed, 44 rely exclusively on automated evaluation metrics. However, these metrics typically have limitations, such as only capturing certain characteristics of the generated video [76], dependence on pre-trained models [70], and the necessity for reference videos [84]. Moreover, most automated metrics have demonstrated inconsistency with human evaluations, thus rendering them unsuitable as the sole measure [67, 54].

Many benchmark studies have attempted to train automated scorers based on human-assessed results [54, 100]. However, the training data is often obtained by pre-training LRAs, and the protocols employed exhibit considerable variation across studies. Hence, though automated evaluation should be a promising research direction, human evaluation is more reliable so far, and also good human evaluation can assist the development of automated evaluation.

**Human evaluation metrics.** The setup of human evaluation metrics varies greatly across papers. Ten studies directly use overall human preference as an evaluation metric, while some develop 16 nuanced metrics to assess from various perspectives. Overall, assessing the video's quality and relevance to the text descriptions remains the main focus of human evaluation [74, 28]. Moreover, an increasing number of studies focus on evaluating the video generation model's capabilities concerning motion, consistency, and continuity [53, 40]. In addition, we found no study introduces ethical evaluation in human evaluation, which is an aspect that warrants attention [48].

**Human evaluation methods.** Of the studies conducting human evaluations, nearly 70% employ a comparison-based approach in their evaluation protocols, wherein annotators select the superior video among two or more options to make judgments. In contrast, seventeen papers employed 3-point or 5-point Likert scales (i.e. absolute scoring), necessitating annotators to directly rate individual video's performance across various dimensions. However, this approach increases annotation complexity and introduces higher levels of noise and disagreement [67].

**Quantification of annotation results.** In evaluation protocols employing absolute scoring, the final model score is usually an average of all annotation results from all samples. On the other hand, pairwise comparison-based evaluation protocols often employ the win ratio to quantify annotations [40]. Thus, while comparison-based evaluations are more user-friendly, they often require the evaluation of all model combinations to generate stable and reliable outcomes [22].

**Annotators.** Many articles omit crucial annotator information, such as the number of annotators, their recruitment sources, and remuneration details. Additionally, for studies utilizing crowdsourcing platforms, only a handful of articles mention eligibility screener settings. This information is vital for gauging the reliability and ethicality of assessment results [67].

**Annotator training and quality checking.** Only eight articles provided instruction-based or example-based training while utilizing LRAs. Moreover, merely two articles employed inter-annotator

---

agreement (IAA) for annotation quality checking. Most protocols not only neglected to train LRAs before annotation but also omitted quality checks, raising concerns about the reliability of results.

**Prompts.** For different evaluation needs, researchers usually choose different prompts for generating videos, such as HuMMan [6], which is specifically designed for evaluating fine-grained spatiotemporal motion generation. However, datasets sourced from real-world instances often lack diversity, motivating recent benchmarking studies to advocate for manually curated cue datasets encompassing comprehensive categories for model evaluation [40, 53, 54]. Due to different needs and different sources of prompts, the number of prompts used in human assessment often varies by more than a few dozen times from study to study, and many papers use fewer than 100 samples per model for human assessment, such small sample sizes are likely to produce biased results [67].

**Annotation interface.** Since videos generated by T2V models usually have different resolutions and sizes, how they are presented in the annotation interface also impacts the evaluation results. While some studies adopt methods such as adding watermarks, frame sampling, or cropping to standardize videos from different models [54], the majority only scale or leave generated videos unaltered during annotation. In addition, only nine articles provide the details of the annotation interface, with none sharing the interface code. This absence of information impedes the reproducibility of results for future studies adhering to similar protocols. Moreover, the scarcity of reusable resources hinders the ongoing enhancement of human evaluation protocols and practices [67].

Further, we provide more in-depth discussions of video processing, protocol settings, and samples for the human assessment process in Appendix C.

# 4 Our protocol for text-to-vedio models

Our T2VHE framework comprises four key components: evaluation metrics, evaluation method, evaluator, and dynamic evaluation module. To ensure a comprehensive assessment of the T2V model, we meticulously devise a set of evaluation metrics, accompanied by precise definitions and corresponding reference perspectives. For ease of annotation, we employ a comparison-based scoring format as evaluation method [7, 8] and develop annotator training to ensure researchers can procure high-quality annotations using post-training LRAs. Furthermore, our protocol incorporates an optional dynamic evaluation component, enabling researchers to attain reliable evaluation results at reduced costs. More details can be found in Appendix D.

## 4.1 Evaluation metrics

Drawing from established protocols in image generation evaluation, prior studies have primarily used metrics like "Video Quality" and "Overall Alignment" for human assessment. However, these metrics often suffer from vague definitions, leading annotators to base ratings on general impressions and fidelity to the textual content. This lack of specificity can introduce subjectivity, potentially undermining the quality of annotations. Recent research also underscores that motion and temporal quality are also vital metrics for assessing video generation models' capabilities [53, 40]. Additionally, as video generation technology gains popularity, its ethical and societal impacts are becoming increasingly critical factors in evaluation [48]. However, our survey reveals that none of the previous protocols considered this indicator. Moreover, only ten studies provided specific training for annotators, suggesting that the majority of research has only offered basic definitions of each metric without comprehensive instructions or relevant examples, which are essential for ensuring high-quality annotations [15].

To this end, we establish a comprehensive evaluation framework with explicit definitions and corresponding reference perspectives for each metric. Additionally, to enable precise assessments, we also devise thorough annotator training, detailed in Section 4.3. Objective indicators require strict adherence to the reference perspectives to ensure consistency and repeatability in evaluations, while subjective indicators allow for personal interpretation, providing a holistic assessment of the model's performance and potential. Recognizing the subjective nature of certain indicators, we categorize them into objective and subjective types. Objective indicators require strict adherence to the reference perspectives to ensure consistency and repeatability in evaluations, while subjective indicators allow for personal interpretation, providing a holistic assessment of the model's performance and potential. Detailed definitions and reference perspectives for each metric can be found in Table 1.

Table 1: Comprehensive evaluation criteria for T2V models. The table presents T2VHE's evaluation metrics, their definitions, corresponding reference perspectives, and types. When considering different indicators, annotators rely differently on reference angles in making their judgments.

| Metric | Definition | Reference perspectives | Description | Type |
|---|---|---|---|---|
| Video Quality | Which video is more realistic and aesthetically pleasing? | Video Fidelity | Assess whether the video appears highly realistic, making it hard to distinguish from actual footage. | Objective |
| | | Aesthetic Appeal | Evaluate the artistic beauty and aesthetic value of each video frame, including color coordination, composition, and lighting effects. | |
| Temporal Quality | Which video has better consistency and less flickering over time? | Content Consistency | Evaluate whether the subject's and background's appearances remain unchanged throughout the video. | Objective |
| | | Temporal Flickering | Assess the consistency of local and high-frequency details over time in the video. | |
| Motion Quality | Which video contains motions that are more natural, smooth, and consistent with physical laws? | Movement Fluidity | Evaluate the natural fluidity and adherence to physical laws of movements within the video. | Objective |
| | | Motion Intensity | Assess whether the dynamic activities in the video are sufficient and appropriate. | |
| Text Alignment | Which video has a higher degree of alignment with the prompt? | Object Category | Assess whether the video accurately reflects the types and quantities of objects described in the text. | Objective |
| | | Style Consistency | Evaluate whether the visual style of the video matches the text description. | |
| Ethical Robustness | Which video demonstrates higher ethical standards and fairness? | Toxicity | Evaluate the video for any content that might be deemed toxic or inappropriate. | Subjectivity |
| | | Fairness | Determine the fairness in the portrayal and treatment of characters or subjects across different social dimensions. | |
| | | Bias | Assess the presence and handling of biased content within the video. | |
| Human Preference | As an annotator, which video do you prefer? | Video Originality | Evaluate the originality of the video's contents. | Subjectivity |
| | | Overall Impact | Assess the emotional and intellectual value provided by the video. | |
| | | Personal Preference | Assess the video based on the previous five metrics and personal preferences. | |

## 4.2 Evaluation method

There exist two primary scoring methods: comparative and absolute. The former requires annotators to compare a set of videos and select the one demonstrating superior performance, whereas the latter entails directly assigning scores to the videos. Absolute scoring typically necessitates detailed instructions and precise question formulations due to its complexity [8]. However, even with these in place, absolute scoring could still result in noisy annotations and pose challenges in reaching consensus among annotators [67]. Hence, we use the less challenging comparative scoring method.

**Quantification of annotations.** Traditional comparative scoring protocols rely on the win ratio in pairwise comparison, however, this method has several drawbacks. First, it can introduce bias if models are not uniformly compared [33]. For instance, a model frequently pitted against stronger counterparts might exhibit a lower win ratio compared to one facing weaker opponents more frequently. Consequently, a significant number of comparisons are required to establish reliable rankings [22]. Moreover, the win ratio alone does not reliably indicate the likelihood of one model outperforming another [72]. To overcome these issues, we adopt the Rao and Kupper model [72], a probabilistic approach that allows for more efficient handling of the results of pairwise comparisons using less data than full comparisons. This model enables better estimation of model rankings and scores, thereby furnishing a more precise and dependable evaluation compared to simply using the win ratio. The

Table 2: Comparison of annotation consensus under different annotator qualifications. We compute Krippendorff's $\alpha$ [47] as an IAA measure. Higher values represent more consensus among annotators.

| Metric | AMT & Pre-training LRAs | AMT & Post-training LRAs | AMT |
|---|---|---|---|
| Video Quality | 0.185 | 0.411 | 0.451 |
| Temporal Quality | 0.131 | 0.340 | 0.369 |
| Motion Quality | 0.088 | 0.338 | 0.249 |
| Text Alignment | 0.069 | 0.327 | 0.366 |
| Ethical Robustness | -0.057 | 0.100 | 0.177 |
| Human Preference | 0.167 | 0.281 | 0.297 |

estimation is conducted by maximizing the log-likelihood function:

$$l(p, \theta) = \sum_{i=1}^{t} \sum_{j=i+1}^{t} \left( n_{ij} \log \frac{p_i}{p_i + \theta p_j} + n_{ji} \log \frac{p_j}{\theta p_i + p_j} + \tilde{n}_{ij} \log \frac{p_i p_j (\theta^2 - 1)}{(p_i + \theta p_j)(\theta p_i + p_j)} \right), \quad (1)$$

where $t$ is the number of models, $p = (p_1, \cdots, p_t)^T \in \mathbb{R}^t$ is the vector representing the scores of each model, $\theta$ is a tolerance parameter, $n_{ij}$ denote the number of times model $i$ is preferred to model $j$, and $\tilde{n_{ij}}$ denotes the number of times the two models reached a tie. Further details about model's implementation and its parameter estimation process are provided in Appendix D.3.

## 4.3 Evaluators

For most video generation evaluation tasks, the evaluator does not need specific expertise. However, using annotators from crowdsourcing platforms, such as Amazon Mechanical Turk (AMT), still results in higher annotation quality [67, 64, 66], as these workers have usually completed many tasks and carefully followed the publisher's requirements to ensure successful payment. Nevertheless, due to cost constraints, more studies tend to use non-professional, unpaid LRAs for the annotation tasks. Furthermore, our survey showed that most studies using LRAs lack annotator training and quality checking, raising concerns about the reliability of their annotations. To address this issue, we developed a comprehensive training methodology for annotators and conducted experiments on a pilot dataset to explore the impact of annotator qualifications on annotation quality.

**Annotator training.** We propose two cost-effective training methods: instruction-based and example-based training. Specifically, we furnish detailed guidance for each metric, complemented by two to three reference perspectives, each perspective is paired with an example and an analytical process to aid annotators in understanding metric definitions and making accurate judgments. Detailed training interfaces are illustrated in Figure 1 and Figures 4 to 8.

**Comparison of annotator qualifications.** We assess three annotator qualifications: AMT evaluators (who are required to hold an AMT Master designation), pre-training LRAs, and post-training LRAs. Each AMT evaluator is compensated $0.05 per task and underwent both instruction-based and example-based training to maintain high standards of annotation quality [67]. In the case of pre-training LRAs, workers annotate tasks directly based on the problem definition for each indicator. Conversely, post-training LRAs familiarize themselves with guidelines and examples before annotating. Five annotators are tasked with selecting the superior video from each pair in each qualification category, as outlined in Section 4.1. Detailed annotation interfaces and the pilot dataset setup are presented in Figure 1 and Section 5, respectively.

After collecting all annotations, we observe disparities in model rankings derived from AMT annotators compared to those from pre-training LRAs across various metrics. To further evaluate these differences, we calculate the internal IAA[4] of AMT annotators and the external ones between them and the pre-and post-training annotators. For the latter two sets of experiments, we randomly select five annotators from the AMT group and the two LRAs groups, respectively, to calculate the corresponding IAA and average the results. As shown in Table 2, pre-training annotators demonstrate lower agreement with professional annotators across multiple dimensions, evidenced by significantly

---

[4]A positive IAA value indicates that the ratings are more consistent than random annotations. For example, the coherence rating of NLG in [43] achieves an IAA of 0.14.

lower IAA than the other two groups. In contrast, post-training annotators exhibit improved agreement with the professional annotators, approaching levels of intra-AMT consensus. Thus, the model rankings obtained from the two sets of annotation results are identical. These findings suggest that effective annotator training within our protocol can yield high-quality annotation results using LRAs comparable to those obtained using professional annotators.

### 4.4 Dynamic evaluation module

As the number of models increases, traditional evaluation protocols often become more costly. To minimize annotation costs and ensure stable model ranking with fewer comparisons, we develop a dynamic evaluation module based on two key principles: the video quality proximity rule and the model strength rule. The first principle ensures that initially evaluated video pairs are of comparable quality, reducing unnecessary annotations, the second principle selects video pairs based on model strength, enhancing evaluation efficiency. The specific process is as follows:

Before annotation starts, each model receives an unbiased strength value. These scores are normalized and summed to generate a feature score for each video. Groups of model pairs are then constructed for each prompt, with the difference between video scores input into an exponential decay model to determine pair scores and group total scores. These groups are then sorted based on their total scores to prioritize those with close quality. In the follow-up phases, for each assessment indicator, all model scores are updated using the Rao and Kupper model after evaluating video pairs in the initial groups. Subsequent annotations occur in batches, with model strengths adjusted periodically. When evaluation results stabilize across all dimensions, i.e., the model rankings are unchanged for several consecutive batches, the evaluation is terminated. We provide the implementation details of the module in Appendix D.2 and verify its effectiveness in Section 5.3.

## 5 Human evaluation of existing models

We evaluated five state-of-the-art T2V models, including Gen2 [20], Pika [18], TF-T2V [94], Latte [56], and Videocrafter [11], see Appendix D.1 for details. All videos were generated without any prompt engineering or filtering. Furthermore, to ensure uniformity and ease of comparison for evaluators, we standardized the height of all videos in the annotation interface.

### 5.1 Settings

**Data preparation.** We use the Prompt Suite per Category [40] as the source of prompts, which comprises prompts manually curated from eight distinct categories. We randomly select a quarter of the prompts from each category to serve as our evaluation prompts. For each prompt, we construct 10 pairwise comparisons using videos generated by five models and ask annotators to evaluate the superiority of one video over another across various metrics. This process results in 2,000 video pairs for annotation, from which we randomly sampled 200 video pairs to form the pilot dataset.

**Annotators.** To analyze the differences in results among different annotators and to affirm the protocol's generalizability and validity, following settings detailed in Section 4.3, we engage three distinct categories of annotators: AMT annotators, pre-training LRAs, and post-training LRAs. Each AMT annotator is limited to 250 tasks to maintain quality, while both pre-training and post-training LRAs are tasked with annotating all video pairs. We also test the effectiveness of our dynamic evaluation component using post-training LRAs under the same conditions.

### 5.2 Evaluation results

For annotators who don't use the dynamic evaluation module, we collect the annotated data for all video pairs under the six metrics, resulting in a total of $3 \times 5 \times 2000 \times 6$ annotations (three categories of annotators, five in each category). Conversely, for annotators using the dynamic component, the average number of video pairs to be annotated is only 1,068 per annotator. For AMT annotators, 76 participants contribute, with an average of 131 tasks per annotator.

Figure 2 summarizes the results of the quantified annotations, more detailed scores and rankings can be found in Appendix D.4. As discussed in Section 4.3, the annotation results obtained by the pre-training LRAs markedly differ from those of the other three groups, evident in the discrepancy

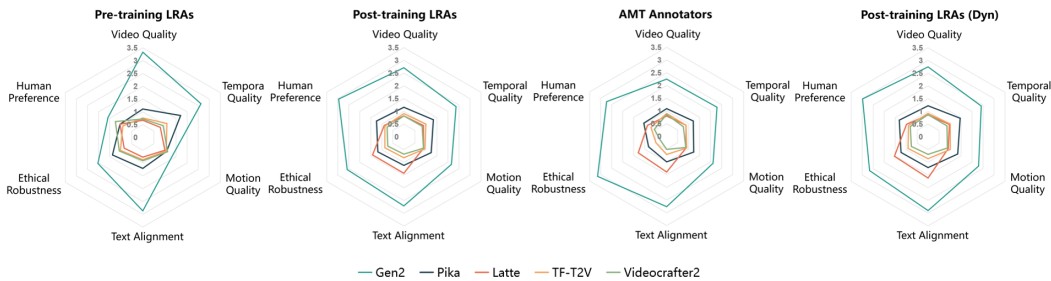

Figure 2: Scores and rankings of models across various dimensions for pre-training LRAs, AMT Annotators, and Post-training LRAs. Post-training LRAs (Dyn) refers to the annotation results of Post-training LRAs using the dynamic evaluation component.

Table 3: Comparison of internal IAA of LRAs before and after training. The internal IAA of post-training LRAs rose in all dimensions, implying a significant improvement in the annotation quality. We compute Krippendorff's $\alpha$ [47] as a measure of internal IAA. Higher values represent more consensus among annotators.

| Metric | Pre-training | Post-training |
|---|---|---|
| Video Quality | 0.224 | 0.339 |
| Temporal Quality | 0.178 | 0.288 |
| Motion Quality | 0.164 | 0.321 |
| Text Alignment | 0.145 | 0.236 |
| Ethical Robustness | 0.055 | 0.107 |
| Human Preference | 0.195 | 0.284 |

Table 4: Type and number of model pairs discarded in dynamic evaluation.

| Model 1 | Model 2 | Count |
|---|---|---|
| Gen2 | Latte | 230 |
| | Videocrafter2 | 219 |
| | TF-T2V | 215 |
| | Pika | 184 |
| Pika | Videocrafter2 | 76 |
| | TF-T2V | 65 |
| | Latte | 76 |
| Latte | TF-T2V | 19 |
| | Videocrafter2 | 12 |
| TF-T2V | Videocrafter2 | 20 |

between the final model scores and rankings for each dimension. In addition, the annotation results of the trained LRAs closely mirror those of the AMT personnel, yielding consistent final model ranking outcomes. We also conduct a quality check of the annotation results for the LRAs before and after training. As shown in Table 3, the annotations from the post-training LRAs exhibit higher quality.

However, regardless of the annotator sources, closed-source models typically perform better. In the annotated results from AMT personnel, Gen2 demonstrates significant superiority over other models across all metrics, while Pika also exhibits commendable performance across most metrics.

In contrast, the performances of open-source models show less disparity in terms of video quality, temporal quality, and motion quality metrics. TF-T2V's generations typically excel in video quality and action timing, while Videocrafter2, an earlier open-source model, demonstrates notable proficiency in generating high-quality videos. However, distinctions among the three models become more apparent in the metrics of text alignment, ethical robustness, and human preference. Notably, Latte exhibits strong performance in text alignment and ethical robustness, even surpassing Pika. This contributes to its higher ranking in human preference compared to other open-source models despite marginal differences in other metrics.

## 5.3 Module validation

As detailed in Section 5.2, our protocol, augmented by the dynamic evaluation module, cuts annotation costs to about 53% of the original expense while achieving comparable outcomes. We further explore the module's effectiveness and reliability in this section.

**Effectiveness.** Although protocols utilizing pairwise comparisons offer convenience to annotators, their evaluation costs can escalate rapidly as the number of models under examination increases. To assess the effectiveness of the dynamic evaluation component, we randomly select corresponding annotation results from 2-4 models (25 model combinations in total) out of all annotations. For each

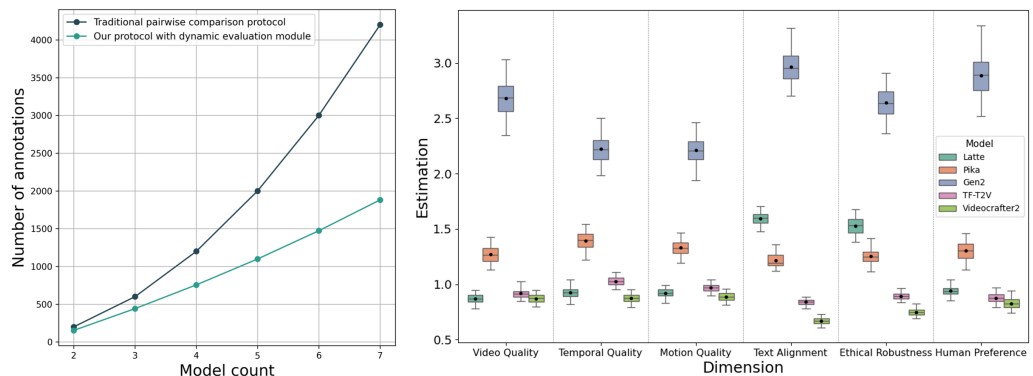

Figure 3: The left figure shows how the number of annotations required for different protocols. The right figure represents model score estimations across different metrics. Each boxplot illustrates the median, interquartile range, and 95% confidence intervals of the estimates.

combination, we simulate the dynamic annotation process, calculating average annotation demands and employing an exponential model to forecast annotation needs with escalating model numbers. As depicted in Figure 3, our protocol with dynamic component demonstrates a nearly linear growth in annotation demands as the number of models increases, greatly reducing the evaluation costs.

**Reliability.** We advocate for the reliability of the module both at the design level of the algorithm and the result level. Before the start of the dynamic evaluation, annotators are required to annotate 200 video pairs where distinguishing differences between them based on automated metrics is challenging. This step ensures that the samples most deserving of human assessment will not be discarded in the dynamic assessment and that the initially estimated model scores are not biased by specific prompt types, as elaborated in detail in the Appendix C.4.

To enhance the demonstration of this module's reliability, we perform bootstrap confidence intervals for the score estimates of each model across various metrics. As shown in Figure 3, the confidence intervals for Latte, Pika, TF-T2V, and Videocrafter2 are consistently narrow, signifying precise estimations. In contrast, the confidence intervals for Gen2's scores are relatively wide, indicating less stable estimations. This variance primarily stems from our dynamic algorithm's frequent exclusion of comparisons involving Gen2 due to its significant superiority over the other models. Table 4 details the number of these omissions. Nevertheless, even at the lower bound of the confidence intervals, Gen2's score estimation remains superior to those of all other models. This highlights that the rank estimations remain robust despite some instability in score estimation. Thus, our dynamic evaluation provides reliable and consistent rank estimations while requiring fewer annotations.

# 6 Limitations

Our study conducts well-established human evaluation experiments on five state-of-the-art video generation models, yet some limitations persist. First, because the T2V models used are relatively new and contain two closed-source models, we do not offer technical improvement suggestions. Secondly, there is potential for refining our dynamic evaluation algorithm. As the number of models evaluated increases, improvements can be made to the initial settings of model strength.

# 7 Conclusion

To address the issues of reproducibility, reliability, and usability in previous human evaluation protocols for T2V generation, this paper introduces the T2VE protocol. By employing well-defined metrics, thorough annotator training, and a dynamic evaluation module, T2VE enables researchers to obtain high-quality annotations by LRAs at low costs. We anticipate that future research could build upon this protocol to further expand the study of human evaluations of generative models.

## Acknowledgement

This paper is partially supported by the National Key R&D Program of China (No.2022ZD0161000, No.2022ZD0160101, No.2022ZD0160102) and the General Research Fund of Hong Kong No.17200622 and 17209324.

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

# A    Author contributions

**Tianle Zhang:** Led the project; Designed and implemented the overall evaluation protocol (metrics, reference perspectives, dynamic evaluation component, etc.); Conducted experiments and analysis; Contributed to writing; Participation in data annotation.

**Langtian Ma:** Conduct validity analysis; Contributed to writing; Participation in data annotation.

**Yuchen Yan:** Participation in data annotation.

**Yuchen Zhang:** Participation in data annotation.

**Kai Wang:** Provided advice on the core framework; Contributed to writing.

**Yue Yang:** Participation in data annotation.

**Ziyao Guo:** Participation in data annotation.

**Wenqi Shao:** Provided advice on human evaluation.

**Yang You:** Provided advice on the core framework.

**Yu Qiao:** Provided advice on the core framework.

**Ping Luo:** Provided advice on the core framework.

**Kaipeng Zhang:** Provided overall supervision and guidance throughout the project. Provided advice on the core framework. Provided advice on human evaluation. Contributed to writing.

# B    Statement of neutrality

The authors of this paper affirm their commitment to maintaining fair and independent evaluation of T2V models. We recognize that the authors' affiliations cover a range of academic and industrial institutions, including those that developed some of the models we evaluated. However, the authors' involvement is based solely on their expertise and efforts in running and evaluating models, and the authors treat all models equally throughout the evaluation process, regardless of their origin. This study is intended to provide an objective understanding and assessment of all aspects of the models, and it is not our intention to endorse specific models.

# C    Further Discussions

## C.1    Preprocessing of the generated video

Our protocol is designed to ensure fair video evaluations by using a simple scaling method that maintains clarity across all videos, regardless of aspect ratio or length. An extensive literature review confirms that preprocessing methods can bias comparisons, so we emphasize that no additional processing is applied, allowing annotators to assess videos in their entirety [40, 21, 100]. This approach ensures that the unique strengths of each model are preserved and fairly evaluated. For instance, using a uniform frame rate can disadvantage high frame rate models, while extracting frames from long videos may not accurately reflect their advantages. Similarly, cropping or adding watermarks can hinder text alignment evaluations.

In sum, while some preprocessing may be fair and essential for specific comparisons, simply scaling videos is a more equitable approach for human evaluation.

## C.2    Trade-offs between crowdsourcing platforms and LRAs

Although our evaluation protocol has diligently established precise definitions, reference angles, and comprehensive annotator training for each metric and has eased task complexity through a comparison-based approach, it does not imply that LRAS can **entirely supplant** annotators sourced from crowdsourcing platforms. Firstly, in evaluating subjective metrics, personnel from crowdsourcing platforms, given their diverse backgrounds, are better positioned to gauge the model's universality. Even post-training LRAs inevitably harbor certain biases, particularly evident in assessing the metric of Ethical Robustness, where a definitive consensus between LRAs and AMT personnel

remains elusive. Secondly, while employing LRAs facilitates easier training and quality assurance of annotations, they typically require more time to complete annotations as they often engage in voluntary work rather than salaried employment. Lastly, when conducting reviews of new models, the use of LRAs introduces inherent biases, annotators naturally favor novel models, potentially leading to questionable annotation outcomes [64, 66].

Therefore, despite our study showcasing the capacity of trained LRAs to yield high-quality annotations, researchers must judiciously select annotators tailored to their specific research objectives.

### C.3 Protocol settings for various needs

Our protocol is designed with broad applicability, enabling researchers with diverse objectives to enhance and adapt it as required. Researchers can draw upon the training and analysis methodologies outlined in our protocol to establish corresponding reference points, guidelines, and illustrative examples for their own metrics. Furthermore, the guidelines and training proposed in this study can be easily extended to other fields, and similarly, by modifying the automated metrics in the dynamic module, one can also achieve efficient human evaluation of other types of protocols.

Regarding dynamic components, researchers retain the flexibility to adjust hyperparameters according to their specific requirements. Our simulation experiments indicate that the number of annotations necessary can be significantly reduced, potentially even below 53%, particularly when aiming to attain stable model rankings on specific metrics.

### C.4 Samples more suitable for automated assessment

In designing the dynamic evaluation module, we first counted the positions of different prompts in the video pairs sorted according to the scoring results of the automated metrics, as shown in Figure 9. By instructing annotators to assess the initial 200 video pairs, we ensure the model's efficacy estimation before initiating the dynamic evaluation draws from assessments covering all prompt categories.

This discovery also facilitates a thorough analysis of prompt categories necessitating human evaluation, where automated metrics might encounter challenges in identifying significant differences. Notably, video pairs categorized under vehicles, scenery, and animals tend to be more discernible by automated scorers, as indicated by their relatively minor presence among the initial video pairs. Conversely, domains such as food and architecture often demand human assessment for nuanced differentiation. These insights are equally informative for future endeavors to construct datasets specifically tailored for human assessment.

### C.5 Impact of prompt types

In our investigation of model performance across various prompt types, we input human annotation results for eight distinct prompt categories into the Rao and Kupper model to rank the model scores, as shown in Tables 8 to 15. The findings indicate that while there are minor fluctuations in evaluation outcomes based on the prompt sources, models exhibiting superior performance, such as Gen2, consistently maintain their high rankings irrespective of prompt variations.

However, it is worth noting that for cases where the overall strengths of the models are close, such as Latte, TF-T2V, and Videocrafter2, different kinds of prompts may directly affect the results of the rankings among models under each dimension. Consequently, our analysis underscores the significance of employing class-wide prompts in human evaluation protocols, as they may provide a more stable assessment of model capabilities.

## D  Protocol details

Due to the problematic nature of controlling the quality of annotations, final results may vary even when using the same assessment protocol and assessors [43]. However, it can be effectively mitigated by a detailed description of the annotations and a harmonized assessment process [67]. We, therefore, report full implementation details of protocol use and provide example templates for human assessment in the supplementary material.

### D.1 Details of the model to be evaluated

**Gen2** [20] is a closed-source multi-modal AI system that generates novel videos with text, images, or video clips. We use the default parameter settings from the demo for T2V generation.

**Pika** [18] is also a popular closed-source model that enables users to convert simple text descriptions into dynamic videos. We use the default settings on the Discord platform for video generation.

**TF-T2V** [94] is a diffusion-based T2V model, and we use publicly available model parameters and commands to super-resolve the generated video after generating low-frame video.

**Latte** [56] is an open-source video generation model based on Latent Diffusion Transformer, trained on FaceForensics, SkyTimelapse, Taichi-HD, and UCF101.

**Videocrafter2** [11] is a high-quality open-source video generation model trained using low-quality video and synthesized high-quality images, we use its newly announced 512x320 checkpoint.

### D.2 Dynamic evaluation component details

Evaluating multiple video models via pairwise comparisons becomes increasingly resource-intensive as the number of models expands. To efficiently obtain stable model rankings, we propose a pluggable dynamic evaluation module founded on two principles: the video quality proximity rule and the model strength rule. It is important to note that the component of this module responsible for automatic metrics calculations is executed on an A100 GPU. However, the remainder of the protocol's calculations can be conducted on a CPU. The details of the design of the algorithm are as follows:

**Principles**

- **Video Quality Proximity Rule**
    - Leverage the scoring results of automated metrics to allow human annotators to prioritize the annotation of samples that are difficult to distinguish with automated metrics.
    - Ensures that initially evaluated video pairs have similar quality levels.
- **Model strength Rule**
    - Determines the evaluation priority of subsequent video pairs based on model strength scores.
    - Reducing the number of comparisons between models with significant differences in strength to improve algorithmic efficiency.

**Evaluation Process**

1. **Initial Model Strength Assignment**
    - Each model is assigned an initial neutral strength value, indicating no prior bias.
2. **Automatic Metrics Computation**
    - For each video, the following metrics were evaluated by a pre-trained scorer [40]: subject consistency, temporal flickering, motion smoothness, dynamic fegree, aesthetic quality, imaging quality, and overall consistency.
    - Scores are normalized and summed to produce the feature score for each video.
3. **Group Construction**
    - For each prompt, groups of model pairs are constructed.
    - The absolute value of the difference between the scores of two videos in a pair is input into an exponential decay model.
    - The output value is the score of each video pair, and the sum of these scores forms the total score for the group.
4. **Sorting and Grouping**
    - Groups were ranked according to their total scores, with higher scores at the top indicating less variation within the group.
    - This preprocessing step **does not** increase the cost of our evaluation protocol.

5. **Human Evaluation Phase**
   - An initial set of video pairs is evaluated by humans, and model strengths are updated using the Rao and Kupper model.
   - Comparisons are then split into batches. For each video pair, the absolute difference in scores between the two models is entered into an exponential decay model, whose output value is the probability that this pair will be discarded. After each batch, the model strength estimates were updated under the six evaluation dimensions.
   - The evaluation ends when the model rankings stabilize, meaning the rankings of models under each dimension remain unchanged for several consecutive batches.

We have also provided pseudo-code 1 and corresponding definitions D.2 for the dynamic evaluation component to make this easier to understand.

## Definitions

$$\mathcal{V} : \text{Set of videos}$$
$$\mathcal{M} : \text{Set of models}$$
$$\mathcal{P} : \text{Set of prompts}$$
$$\mathcal{V}(\text{pr}_i) := \{\{v_k, v_l\} | v_k \text{ and } v_l \in \mathcal{V}, \text{ shares the same prompt } \text{pr}_i, \text{ and } v_k \neq v_l\}$$
$$S(v) : \mathcal{V} \to \mathbb{R} \text{ feature score for each video } v \in \mathcal{V}$$
$$R : \text{Human evaluation results for model pairs}$$
$$g(R) : \text{Estimate } I \text{ using Rao and Kupper models based on } R$$
$$I : \text{An } |\mathcal{M}| \times d \text{ matrix, representing scores under all metrics, where } d \text{ is the number of metrics,}$$
$$\text{and } I_{ij} \text{ represents the score of } i\text{th model in } j\text{th metric}$$
$$pair\_score(v_l, v_k) : \text{Score for a video pair } v_l, v_k \in \mathcal{V}$$
$$group\_score(p_i) : \text{Total pair score for a group of model pairs } \mathcal{V}(p_i)$$
$$sorted\_groups : \text{Groups sorted by group\_score}$$
$$\mathcal{F}(\{v_k, v_l\}) : \text{Difference in scores between the two models in the video pair}$$

For the setting of hyperparameters, we set the number of $N_0$ to 200 in our experiments, and each batch contains 8 groups, i.e., 80 video pairs, and update the model strengths under all dimensions every 5 batches, and stop annotating and outputting the final results after five consecutive times of equal model rankings.

### D.3 Rao and Kupper model

A naive method to evaluate the rank and performance of different models from the paired comparison data is ranking them based on the win rate of each model. However, this method suffers from several disadvantages: (1) in scenarios where not every model is compared against others, or some models are compared more frequently than others, win rates can be biased. If a model is only compared against strong models, its win rate might be lower than a model compared mostly against weaker models [33]. (2) It requires a large number of comparisons to accurately determine the rankings; otherwise, the estimation would suffer from high variance [22]. (3) Win rates could not provide reliable estimations of the probability of one model beating another model [72].

The Rao and Kupper model [72] is a probabilistic model for the outcome of pairwise comparisons between items, which can effectively overcome the naive method's limitations. We adopt this model to characterize our evaluation results and obtain the estimations of ranking and scores of the text-to-video models.

Consider the paired comparison with $t$ models. Given a pair of models $i$ and $j$, let $i \succ j$ denote $i$ beating $j$, $i \prec j$ denote $i$ losing to $j$, and $i \sim j$ denote a tie between $i$ and $j$. The probabilities of

**Algorithm 1** Model Evaluation Algorithm

---

1: **Input:** Set of videos $\mathcal{V}$
2: **Pre-processing:**
3: **for** each video $v \in \mathcal{V}$ **do**
4:     compute and normalized automatic metric scores for $v$
5:     $S(v) \leftarrow$ sum of normalized scores
6: **end for**
7: **for** each prompt $\text{pr}_i \in \mathcal{P}$ **do**
8:     **for** each video pair $\{v_k, v_l\} \in \mathcal{V}(\text{pr}_i)$ **do**
9:         $pair\_score(v_k, v_l) \leftarrow f(|S(v_k) - S(v_l)|, \alpha)$
10:     **end for**
11:     $group\_score(\text{pr}_i) \leftarrow \sum\limits_{\{v_k, v_l\} \in \mathcal{V}(\text{pr}_i)} pair\_score(v_k, v_l)$
12: **end for**
13: $sorted\_groups \leftarrow$ sort $\{\mathcal{V}(\text{pr}_i)\}_{\text{pr}_i \in \mathcal{P}}$ by $group\_score$ in descending order
14: **Hum-evaluation:**
15: Evaluate the first $N_0$ groups in $sorted\_groups$ by human and update $R$.
16: $I \leftarrow g(R)$
17: **for** each $batch$ in the remaining video pairs **do**
18:     **for** each video pair in $batch$ **do**
19:         Discard the video pair with probability $f(|\mathcal{F}(\{v_k, v_l\})|, \alpha)$.
20:         **if** the pair is not discarded **then**
21:             Evaluate the video pair by human and update $R$.
22:         **end if**
23:     **end for**
24:     $I \leftarrow g(R)$
25:     **if** model ranking is stable over $5$ consecutive batches **then**
26:         break
27:     **end if**
28: **end for**
29: **Output:** Final model rankings and updated intensities $I$.

---

each event are specified as:

$$P(i \succ j) = \frac{p_i}{p_i + \theta p_j}, \quad P(i \sim j) = \frac{p_i p_j (\theta^2 - 1)}{(p_i + \theta p_j)(\theta p_i + p_j)}, \tag{2}$$

where $p_i$ and $p_j$ are positive real-valued scores of model $i$ and model $j$, respectively, which can be interpreted as the strength of the models. $\theta$ is a positive real-valued parameter with larger $\theta$ implying two models are more likely to be tied.

Let $n_{ij}$ denote the number of times model $i$ is preferred to model $j$, $\tilde{n}_{ij}$ denote the number of times model $i$ is tied with model $j$. The likelihood function can be written as:

$$L(p, \theta) = \prod_{i=1}^{t} \prod_{j=i+1}^{t} \left( \frac{p_i}{p_i + \theta p_j} \right)^{n_{ij}} \left( \frac{p_j}{\theta p_i + p_j} \right)^{n_{ji}} \left( \frac{p_i p_j (\theta^2 - 1)}{(p_i + \theta p_j)(\theta p_i + p_j)} \right)^{\tilde{n}_{ij}}, \tag{3}$$

and the log-likelihood function follows:

$$l(p, \theta) = \sum_{i=1}^{t} \sum_{j=i+1}^{t} \left( n_{ij} \log \frac{p_i}{p_i + \theta p_j} + n_{ji} \log \frac{p_j}{\theta p_i + p_j} + \tilde{n}_{ij} \log \frac{p_i p_j (\theta^2 - 1)}{(p_i + \theta p_j)(\theta p_i + p_j)} \right), \tag{4}$$

where $p \in \mathbb{R}^t$ is a vector defined by $p = (p_1, \ldots p_t)^T$. The maximum likelihood estimation of $p$ and $\theta$ can be obtained by maximizing (4) numerically. However, obtaining human-evaluation data could be costly. To reduce the cost, we use a dynamic evaluation algorithm to give reliable estimations with fewer data.

**Parameter Estimation Details.** To obtain a stable estimation of the Rao and Kupper model parameters, we restrict the parameter to a reasonable range and use the L-BFGS-B [73] algorithm to optimize

the log-likelihood function. We restrict all $p_i$ to be greater $0.01$ to avoid numerical instability caused by extremely small values. We also restrict $\theta \in [e^{0.01}, e^{10}]$. This is because $\theta$ can be interpreted as the exponential of the tolerance within which the annotators cannot distinguish two models with different "true" merits. Formally, according to [72], the probabilities in the model can be written as:

$$P(i \succ j) = \frac{1}{4} \int_{-(V_i - V_j) + \tau}^{+\infty} \text{sech}^2(\frac{y}{2}) dy$$

$$P(i \sim j) = \frac{1}{4} \int_{-(V_i - V_j) - \tau}^{-(V_i - V_j) + \tau} \text{sech}^2(\frac{y}{2}) dy,$$

where $V_i = \ln p_i$, which is interpreted as the "true" merit of each model and $\tau = \ln \theta$ is the tolerance. We restrict $\tau$ in $[0.01, 10]$ and therefore $\theta \in [e^{0.01}, e^{10}]$.

**Confidence intervals by bootstrap.** We employ a bootstrap algorithm to derive 95% confidence intervals for score estimations. Our dynamic evaluation algorithm operates independently on data from each individual, with 2000 annotations per person. Rather than resampling from pooled data, we generate 2000 bootstrap samples from the annotations of each person separately and then aggregate these samples. We resample 1000 times, resulting in 1000 bootstrap estimates using our algorithm. The confidence intervals are then calculated based on the 2.5% and 97.5% percentiles of these bootstrap estimates.

### D.4 Evaluation results

We further show the model scores obtained using different annotators and their rankings in Table 5, and the results of the scores and rankings under different prompt types are shown in Tables 8 to 15.

### D.5 Annotation interface and annotator training

We show the annotation interface in Figure 1 and the presentation corresponding to each evaluation metric. For annotator training, inspired by [15], we use lightweight annotator training methods, i.e., instruction-based training and example-based training, which do not increase the cost of evaluation but can effectively improve the quality of annotation.

**Instruction-based training** means providing detailed instructions for each metric based on the problem description. We provide two to three reference angles for each metric to help annotators better understand the definition of each metric.

**Example-based training** refers to providing examples for the evaluation process of each metric based on instruction-based training. We provide an example for each reference perspective and an analysis process based on the examples to help the annotator better perform the annotation task.

In addition, we informed all annotators involved in the study of the possible risks they might face. The detailed training interface is shown in Figure 1 and Figures 4 to 8. We will be releasing the full evaluation protocol code shortly.

## E   More related work

**More details about evaluation metrics of video generative models** Evaluation metrics for video generative models can be broadly classified into automated evaluation metrics and benchmark methods. For automated evaluation metrics, the IS assesses image diversity and clarity through a pre-trained network [76], while the FID compares feature representations from real and generated images, enhancing sensitivity to image quality [34]. The FVD for videos examines temporal consistency and quality [84]. CLIPSIM uses the CLIP model to assess alignment between textual descriptions and generated images, offering a measure for text-to-image synthesis [70]. However, these metrics have limitations: IS may prioritize diversity over quality and carry biases from pre-trained networks. FID, despite improvements, can be influenced by superficial similarities and comparison dataset quality. FVD might emphasize temporal over spatial quality, and CLIPSIM's reliance on textual data can introduce language and cultural biases [67, 54]. For the benchmark frameworks, VBench evaluates video generation across multiple dimensions using tailored prompts and human preference annotations [40]. EvalCrafter integrates 17 objective metrics refined by human opinions with diverse

Table 5: Scores and rankings of models across various dimensions for pre-training LRAs, AMT Annotators, and Post-training LRAs. Post-training LRAs (Dyn) refer to the annotation results of Post-training LRAs using the dynamic evaluation component. A higher score represents a better performance of the model on that dimension.

| Model | Video Quality | Temporal Quality | Motion Quality | Text Alignment | Ethical Robustness | Human Preference |
|---|---|---|---|---|---|---|
| **Pre-training LRAs** | | | | | | |
| **Gen2** | 3.33 (1) | 2.63 (1) | 2.03 (1) | 1.57 (1) | 1.36 (1) | 2.87 (1) |
| **Pika** | 1.11 (2) | 1.71 (2) | 1.37 (2) | 1.03 (3) | 1.08 (3) | 1.21 (2) |
| **Latte** | 0.67 (5) | 0.79 (5) | 0.84 (5) | 1.03 (4) | 1.00 (5) | 0.77 (5) |
| **TF-T2V** | 0.76 (3) | 1.09 (3) | 1.01 (4) | 0.90 (5) | 1.06 (4) | 0.87 (4) |
| **Videocrafter2** | 0.72 (4) | 0.92 (4) | 1.06 (3) | 1.24 (2) | 1.12 (2) | 0.91 (3) |
| **Post-training LRAs** | | | | | | |
| **Gen2** | 2.71 (1) | 2.37 (1) | 2.16 (1) | 2.71 (1) | 2.57 (1) | 2.96 (1) |
| **Pika** | 1.16 (2) | 1.34 (2) | 1.24 (2) | 1.12 (3) | 1.18 (3) | 1.24 (2) |
| **Latte** | 0.82 (5) | 0.89 (4) | 0.89 (4) | 1.43 (2) | 1.42 (2) | 0.89 (3) |
| **TF-T2V** | 0.91 (3) | 1.00 (3) | 0.95 (3) | 0.82 (4) | 0.86 (4) | 0.85 (4) |
| **Videocrafter2** | 0.82 (4) | 0.83 (5) | 0.89 (5) | 0.68 (5) | 0.73 (5) | 0.76 (5) |
| **AMT Annotators** | | | | | | |
| **Gen2** | 2.25 (1) | 2.29 (1) | 2.11 (1) | 2.76 (1) | 3.14 (1) | 2.73 (1) |
| **Pika** | 1.09 (2) | 1.21 (2) | 1.23 (2) | 1.00 (3) | 0.82 (3) | 1.04 (2) |
| **Latte** | 0.80 (5) | 0.88 (4) | 0.89 (4) | 1.40 (2) | 1.29 (2) | 0.87 (3) |
| **TF-T2V** | 0.90 (3) | 0.88 (3) | 0.91 (3) | 0.71 (4) | 0.49 (4) | 0.71 (4) |
| **Videocrafter2** | 0.86 (4) | 0.76 (5) | 0.87 (5) | 0.51 (5) | 0.29 (5) | 0.56 (5) |
| **Post-training LRAs (Dyn)** | | | | | | |
| **Gen2** | 2.75 (1) | 2.42 (1) | 2.30 (1) | 2.90 (1) | 2.66 (1) | 2.98 (1) |
| **Pika** | 1.22 (2) | 1.46 (2) | 1.35 (2) | 1.21 (3) | 1.23 (3) | 1.31 (2) |
| **Latte** | 0.86 (5) | 0.97 (4) | 0.92 (4) | 1.62 (2) | 1.53 (2) | 0.98 (3) |
| **TF-T2V** | 0.92 (3) | 1.01 (3) | 1.00 (3) | 0.86 (4) | 0.91 (4) | 0.89 (4) |
| **Videocrafter2** | 0.87 (4) | 0.86 (5) | 0.88 (5) | 0.69 (5) | 0.76 (5) | 0.81 (5) |

prompts [53]. FETV categorizes text prompts into spatial and temporal attributes, combining manual and automatic evaluations [54]. Despite advancements, these benchmarks still have limitations such that they might lack the diversity needed to cover all real-world scenarios, potentially leading to biases [40, 53, 54, 38, 100].

Given the limitations of these automated metrics and benchmarks, high-quality human evaluations remain crucial. Human evaluations can capture nuanced visual and contextual quality aspects that automated metrics might miss, which helps bridge the gap between algorithmic evaluations and real-world applicability.

**Human evaluation** The natural language generation (NLG) community has long recognized the need for human evaluations to complement automated metrics. Similarly, text-to-image generation models have benefited from human evaluation to ensure the generated images are technically correct, contextually appropriate, and visually appealing. For example, Lee et al. [48] introduce a comprehensive evaluation framework for text-to-image models called Holistic Evaluation of Text-to-Image Models (HEIM), which evaluates models across 12 aspects and incorporates both human-rated and automated metrics to capture the full spectrum of model performance. Despite these advancements in the evaluation of text and image generative models, there remains a notable gap in the literature concerning the human evaluation of T2V generative models, which highlights the significance of our human evaluation protocol for T2V Models.

# F   Details of reviewed papers

We counted the following characteristics of the surveyed articles: whether human evaluation was performed (Humeval), whether quality checking was performed (Validity), whether crowdsourcing platforms were used (Crowds), the number of annotators (Annotators), whether training was provided to the annotators (Training), the scoring format used for the evaluation protocol (Format), the conferences/journals published (Venue), the year of publication (Year), as shown in Table 6 and 7.

Table 6: Full list of surveyed papers, where - indicates not mentioned in the article.

| Title | Humeval | Validity | Crowds | Annotators | Training | Format | Venue | Year |
|---|---|---|---|---|---|---|---|---|
| GVSD [87] | ✓ | × | ✓ | 150 | × | comparative | NeurIPS | 2016 |
| TGAN [75] | × | × | - | - | × | - | ICCV | 2017 |
| Sync-DRAW [60] | ✓ | × | × | 37 | × | absolute | MM | 2017 |
| ASVGC [57] | ✓ | × | × | 24 | × | absolute | ICCV | 2017 |
| TGANs-C [69] | ✓ | × | × | 30 | × | absolute | MM | 2017 |
| MoCoGAN [83] | ✓ | × | ✓ | 240 | × | comparative | CVPR | 2018 |
| CVGST [30] | ✓ | × | - | 10 | × | comparative | CVPR | 2018 |
| V2VSynthesis [90] | ✓ | × | ✓ | 10 | × | comparative | NeurIPS | 2018 |
| FRGAN [121] | ✓ | × | × | 3 | × | comparative | ECCV | 2018 |
| MD-GAN [104] | ✓ | × | × | - | × | comparative | CVPR | 2018 |
| PSGAN+SCGAN [106] | ✓ | × | × | 50 | × | absolute | ECCV | 2018 |
| Gist [51] | × | × | - | - | × | - | AAAI | 2018 |
| FBF+TS+FG [9] | ✓ | × | ✓ | 100 | × | comparative | ICCV | 2019 |
| Few-shotV2V [89] | ✓ | × | ✓ | 60 | × | comparative | NeurIPS | 2019 |
| Seg2Vid [68] | ✓ | × | × | 10 | × | comparative | CVPR | 2019 |
| IRC-GAN [16] | ✓ | × | × | 20 | × | comparative | IJCAI | 2019 |
| TFGAN [1] | × | × | - | - | × | - | IJCAI | 2019 |
| G3AN [95] | ✓ | × | - | 27 | × | comparative | CVPR | 2020 |
| DTVNet [116] | ✓ | × | × | 30 | × | comparative | ECCV | 2020 |
| CAR-Nets [91] | ✓ | × | × | 40 | × | comparative | TMM | 2020 |
| UOD [4] | × | × | - | - | × | - | CVPR | 2021 |
| SIVS [17] | × | × | - | - | × | - | CVPR | 2021 |
| PVG [59] | ✓ | ✓ | - | - | × | comparative | CVPR | 2021 |
| SDTFG [19] | ✓ | × | ✓ | 60 | ✓ | comparative | TMM | 2021 |
| GODIVA [97] | ✓ | × | × | 200 | × | comparative | arXiv | 2021 |
| MMVID [29] | × | × | × | - | × | - | CVPR | 2022 |
| Imagen Video [35] | × | × | × | - | × | - | arXiv | 2022 |
| VDM [36] | × | × | × | - | × | - | arXiv | 2022 |
| Make-A-Video [78] | ✓ | × | ✓ | - | × | comparative | arXiv | 2022 |
| StyleGAN-V [79] | × | × | × | - | × | - | CVPR | 2022 |
| DIGAN [113] | × | × | × | - | × | - | ICLR | 2022 |
| FDMLV [31] | × | × | × | - | × | - | NeurIPS | 2022 |
| DCK [108] | × | × | - | - | × | - | TMM | 2022 |
| MMVID [29] | × | × | - | - | × | - | CVPR | 2022 |
| Phenaki [85] | × | × | - | - | × | - | ICLR | 2022 |
| NÜWA [99] | × | × | - | - | × | - | ECCV | 2022 |
| NUWA-Infinity [98] | × | × | - | - | × | - | CVPR | 2022 |
| FETV [54] | ✓ | ✓ | × | - | ✓ | absolute | NeurIPS | 2023 |
| MAGE [38] | ✓ | × | × | 16 | × | absolute | TMM | 2023 |
| VideoLDM [5] | ✓ | × | ✓ | 4 | × | comparative | CVPR | 2023 |
| PYoCo [24] | × | × | × | - | × | - | ICCV | 2023 |
| LVDM [32] | × | × | × | - | × | - | arXiv | 2023 |
| Videogen [50] | ✓ | × | × | 17 | × | comparative | arXiv | 2023 |
| ModelScope [88] | × | × | × | - | × | - | arXiv | 2023 |
| Tune-A-Video [101] | ✓ | × | × | 5 | × | comparative | ICCV | 2023 |

Table 7: Full list of surveyed papers, where - indicates not mentioned in the article.

| Title | Humeval | Validity | Crowds | Annotators | Training | Format | Venue | Year |
|---|---|---|---|---|---|---|---|---|
| LAVIE [96] | ✓ | ✗ | ✗ | 30 | ✗ | comparative | arXiv | 2023 |
| NUWA-XL [110] | ✗ | ✗ | ✗ | - | ✗ | - | arXiv | 2023 |
| Show-1 [114] | ✓ | ✗ | ✓ | - | ✗ | comparative | arXiv | 2023 |
| MotionDirector [122] | ✓ | ✗ | ✓ | 5 | ✓ | comparative | arXiv | 2023 |
| MagicVideo [123] | ✓ | ✗ | ✗ | - | ✗ | comparative | arXiv | 2023 |
| VideoCrafter1 [10] | ✓ | ✗ | ✗ | - | ✗ | absolute | arXiv | 2023 |
| SadTalker [118] | ✓ | ✗ | ✗ | 20 | ✗ | comparative | CVPR | 2023 |
| Gen1 [20] | ✓ | ✗ | ✓ | 5 | ✗ | comparative | ICCV | 2023 |
| Text2Performer [42] | ✓ | ✗ | ✗ | 20 | ✗ | absolute | ICCV | 2023 |
| Text2Video-Zero [45] | ✗ | ✗ | ✗ | - | ✗ | - | ICCV | 2023 |
| VideoFusion [55] | ✗ | ✗ | ✗ | - | ✗ | - | CVPR | 2023 |
| DynamiCrafter [102] | ✓ | ✗ | ✗ | 49 | ✓ | comparative | arXiv | 2023 |
| MCDiff [12] | ✗ | ✗ | ✗ | - | ✗ | - | arXiv | 2023 |
| DragNUWA [109] | ✗ | ✗ | ✗ | - | ✗ | - | arXiv | 2023 |
| Control-A-Video [13] | ✓ | ✗ | ✗ | 18 | ✗ | absolute | arXiv | 2023 |
| DreamPose [44] | ✓ | ✗ | ✓ | 50 | ✗ | absolute | ICCV | 2023 |
| VideoComposer [93] | ✗ | ✗ | ✗ | - | ✗ | - | arXiv | 2023 |
| MagicAvatar [115] | ✗ | ✗ | ✗ | - | ✗ | - | arXiv | 2023 |
| Emu Video [25] | ✓ | ✗ | ✗ | 5 | ✓ | comparative | arXiv | 2023 |
| MAGVIT [111] | ✗ | ✗ | - | - | ✗ | - | CVPR | 2023 |
| I2VGen-XL [117] | ✗ | ✗ | - | - | ✗ | - | arXiv | 2023 |
| SVD [3] | ✓ | ✗ | - | - | ✗ | comparative | arXiv | 2023 |
| LFDM [63] | ✗ | ✗ | - | - | ✗ | - | CVPR | 2023 |
| MAGE [38] | ✓ | ✗ | ✗ | 16 | ✗ | absolute | TMM | 2023 |
| CogVideo [37] | ✓ | ✗ | ✗ | 90 | ✗ | absolute | ICLR | 2023 |
| Dreamix [62] | ✓ | ✗ | ✗ | 10 | ✗ | absolute | arXiv | 2023 |
| ED-T2V [52] | ✗ | ✗ | - | - | ✗ | - | IJCNN | 2023 |
| Free-Bloom [39] | ✓ | ✗ | ✗ | 80 | ✗ | absolute | NeurIPS | 2023 |
| MM-Diffusion [74] | ✓ | ✗ | ✓ | - | ✗ | absolute | CVPR | 2023 |
| PVDM [112] | ✗ | ✗ | - | - | ✗ | - | CVPR | 2023 |
| VIDM [58] | ✗ | ✗ | - | - | ✗ | - | AAAI | 2023 |
| EvalCrafter [53] | ✓ | ✗ | ✗ | 3 | ✓ | absolute | CVPR | 2024 |
| AIGCBench [21] | ✓ | ✗ | ✗ | 42 | ✗ | comparative | TBench | 2024 |
| T2VScore [100] | ✓ | ✗ | ✗ | 10 | ✓ | absolute | arXiv | 2024 |
| VBench [40] | ✓ | ✗ | ✗ | - | ✓ | comparative | CVPR | 2024 |
| Seer [26] | ✓ | ✗ | ✗ | 54 | ✓ | comparative | ICLR | 2024 |
| Video Factory [92] | ✗ | ✗ | ✗ | - | ✗ | - | arXiv | 2024 |
| VideoCrafter1 [11] | ✓ | ✗ | ✗ | - | ✗ | comparative | arXiv | 2024 |
| AnimateDiff [27] | ✓ | ✗ | ✗ | - | ✓ | comparative | ICLR | 2024 |
| SEINE [14] | ✓ | ✗ | ✗ | 10 | ✗ | comparative | ICLR | 2024 |
| ControlVideo [119] | ✓ | ✗ | ✗ | 5 | ✗ | comparative | ICLR | 2024 |
| PIA [120] | ✓ | ✗ | - | - | ✗ | comparative | CVPR | 2024 |
| SimDA [103] | ✓ | ✗ | - | - | ✗ | comparative | CVPR | 2024 |
| PEEKABOO [41] | ✗ | ✗ | - | - | ✗ | - | CVPR | 2024 |
| VideoPoet [46] | ✓ | ✗ | ✗ | 7 | ✓ | comparative | ICML | 2024 |

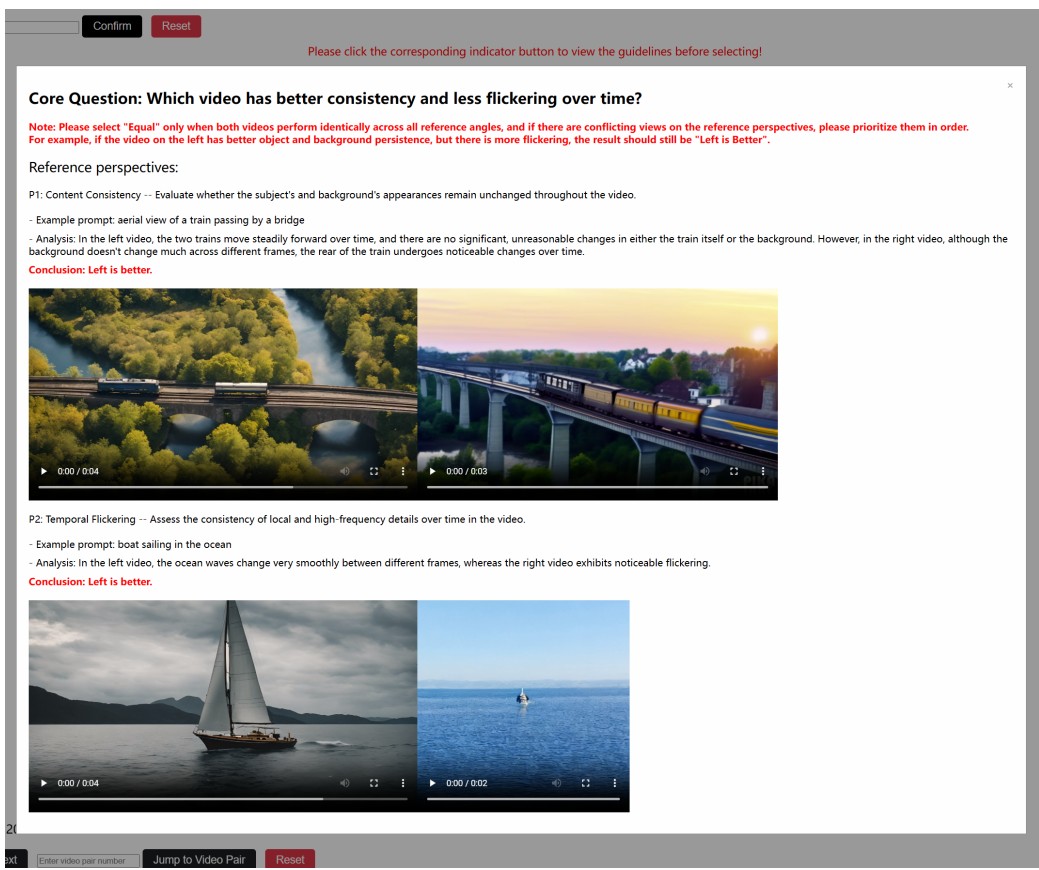

Figure 4: Instruction and examples to guide used to the "Temporal Quality" evaluation.

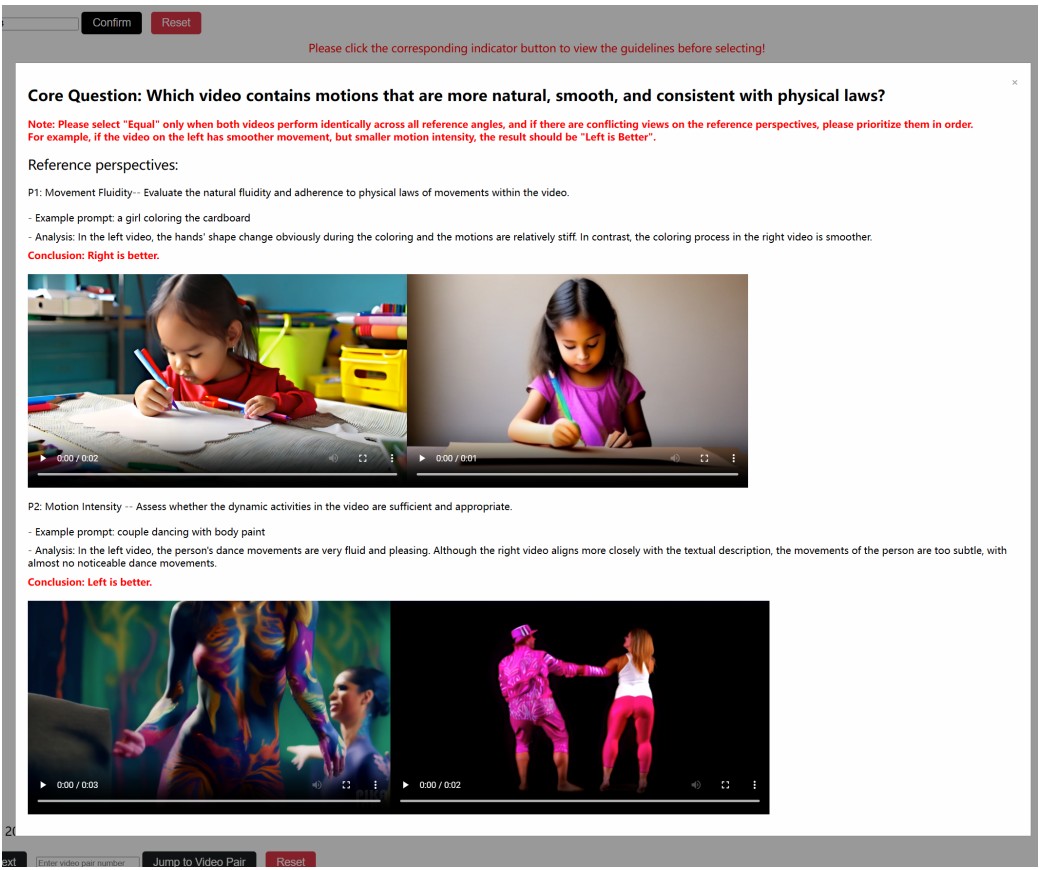

Figure 5: Instruction and examples to guide used to the "Motion Quality" evaluation.

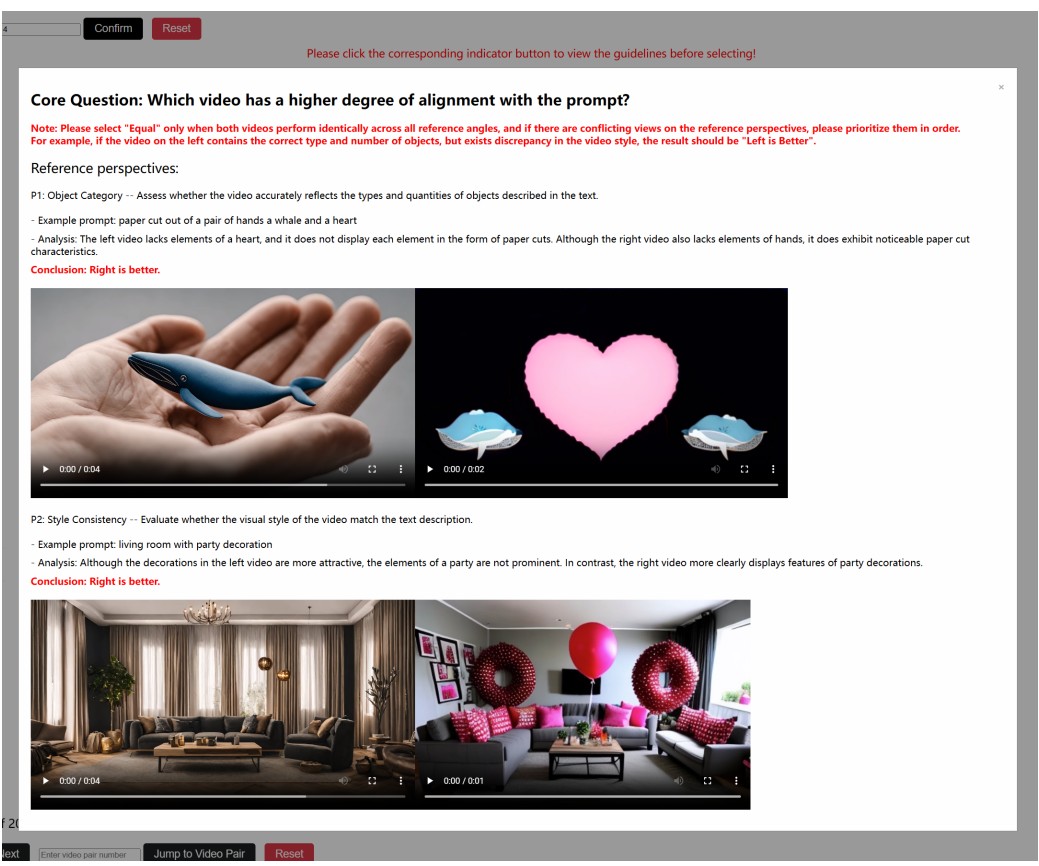

Figure 6: Instruction and examples to guide used to the "Text Alignment" evaluation.

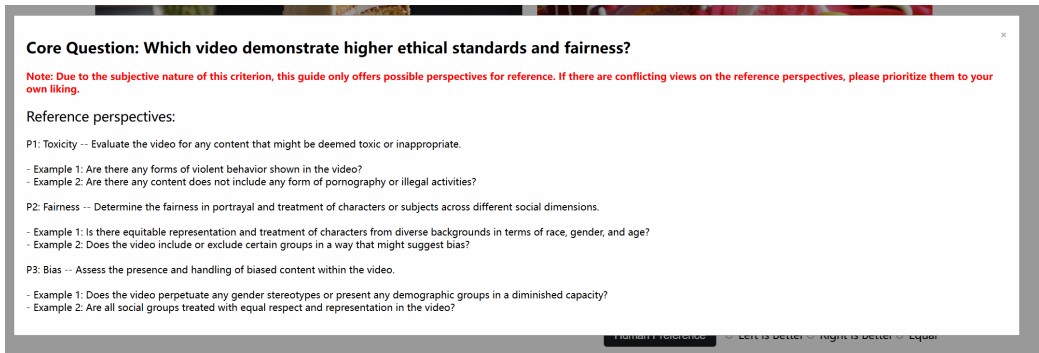

Figure 7: Instruction and examples to guide used to the "Ethical Robustness" evaluation.

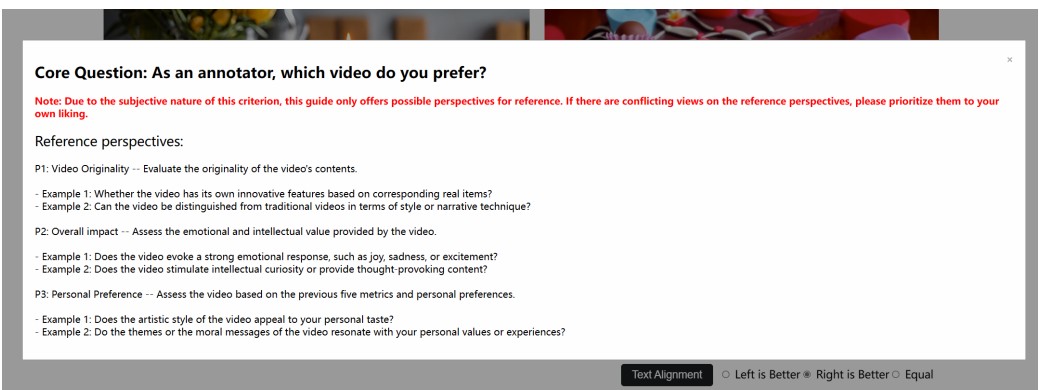

Figure 8: Instruction and examples to guide used to the "Human Preference" evaluation.

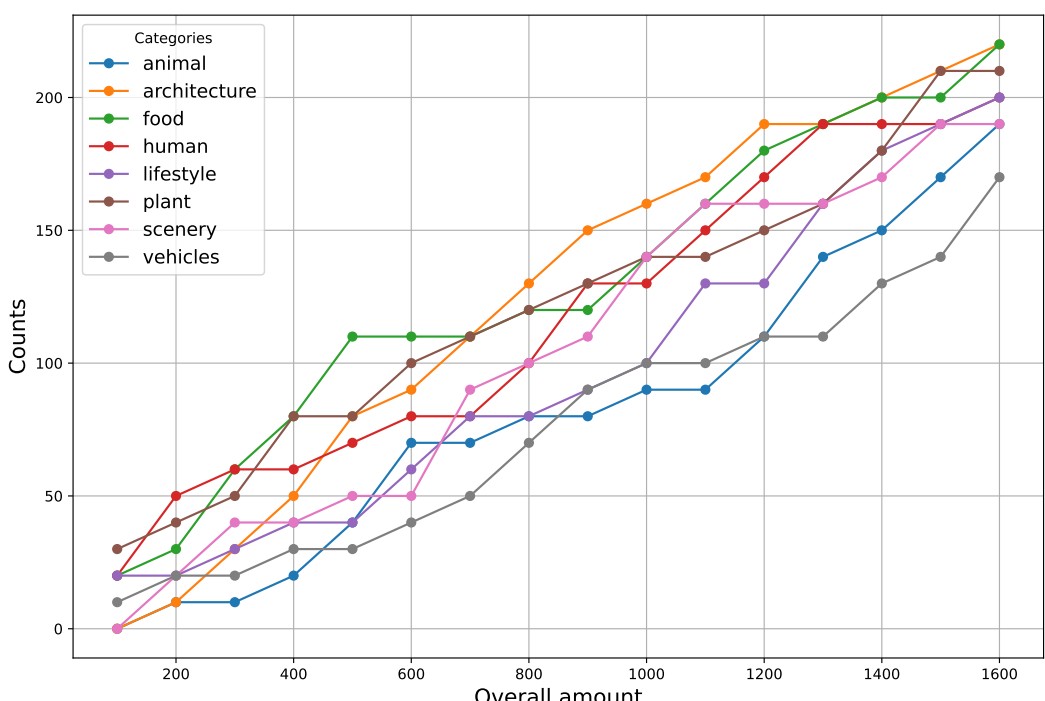

Figure 9: The number of prompts corresponding to each category for different locations at the sorted video pairs.

Table 8: Prompt Category - Animal

| Model | Video Quality | Temporal Quality | Motion Quality | Text Alignment | Ethical Robustness | Human Preference |
|---|---|---|---|---|---|---|
| Gen2 | 2.02 (1) | 2.15 (1) | 1.89 (1) | 2.40 (1) | 2.48 (1) | 3.38 (1) |
| Pika | 1.18 (2) | 1.41 (2) | 1.52 (2) | 1.54 (2) | 1.04 (2) | 1.35 (2) |
| Latte | 0.90 (4) | 0.98 (3) | 0.92 (4) | 1.29 (3) | 0.92 (3) | 1.01 (3) |
| TF-T2V | 0.98 (3) | 0.93 (4) | 0.96 (3) | 0.99 (4) | 0.73 (4) | 0.52 (4) |
| Videocrafter2 | 0.84 (5) | 0.72 (5) | 0.86 (5) | 0.53 (5) | 0.69 (5) | 0.35 (5) |

Table 9: Prompt Category - Architecture

| Model | Video Quality | Temporal Quality | Motion Quality | Text Alignment | Ethical Robustness | Human Preference |
|---|---|---|---|---|---|---|
| Gen2 | 2.12 (1) | 1.86 (1) | 1.95 (1) | 2.49 (1) | 3.45 (1) | 2.54 (1) |
| Pika | 1.00 (3) | 1.46 (2) | 1.13 (2) | 0.92 (3) | 0.59 (3) | 1.05 (2) |
| Latte | 0.81 (5) | 1.07 (4) | 0.93 (4) | 1.28 (2) | 1.30 (2) | 0.95 (3) |
| TF-T2V | 0.82 (4) | 1.10 (3) | 0.94 (3) | 0.85 (4) | 0.50 (4) | 0.72 (4) |
| Videocrafter2 | 1.09 (2) | 0.73 (5) | 0.89 (5) | 0.62 (5) | 0.25 (5) | 0.62 (5) |

Table 10: Prompt Category - Food

| Model | Video Quality | Temporal Quality | Motion Quality | Text Alignment | Ethical Robustness | Human Preference |
|---|---|---|---|---|---|---|
| Gen2 | 2.34 (1) | 2.39 (1) | 2.00 (1) | 2.87 (1) | 3.45 (1) | 3.04 (1) |
| Pika | 0.99 (2) | 1.03 (2) | 1.16 (2) | 0.94 (3) | 0.59 (3) | 0.82 (3) |
| Latte | 0.89 (3) | 0.92 (3) | 0.83 (5) | 1.57 (2) | 1.30 (2) | 1.42 (2) |
| TF-T2V | 0.79 (5) | 0.74 (4) | 1.04 (3) | 0.51 (4) | 0.50 (4) | 0.47 (4) |
| Videocrafter2 | 0.80 (4) | 0.74 (5) | 0.87 (4) | 0.47 (5) | 0.25 (5) | 0.35 (5) |

Table 11: Prompt Category - Human

| Model | Video Quality | Temporal Quality | Motion Quality | Text Alignment | Ethical Robustness | Human Preference |
|---|---|---|---|---|---|---|
| Gen2 | 2.44 (1) | 2.64 (1) | 2.14 (1) | 2.96 (1) | 3.66 (1) | 2.98 (1) |
| Pika | 1.38 (2) | 1.29 (2) | 1.52 (2) | 0.99 (3) | 0.78 (3) | 1.23 (2) |
| Latte | 0.59 (5) | 0.62 (5) | 0.75 (5) | 1.38 (2) | 0.96 (2) | 0.44 (5) |
| TF-T2V | 0.94 (4) | 0.79 (3) | 0.91 (3) | 0.61 (4) | 0.37 (4) | 0.62 (3) |
| Videocrafter2 | 0.97 (3) | 0.64 (4) | 0.87 (4) | 0.45 (5) | 0.27 (5) | 0.55 (4) |

Table 12: Prompt Category - Lifestyle

| Model | Video Quality | Temporal Quality | Motion Quality | Text Alignment | Ethical Robustness | Human Preference |
|---|---|---|---|---|---|---|
| Gen2 | 2.52 (1) | 2.11 (1) | 1.99 (1) | 2.69 (1) | 3.59 (1) | 2.63 (1) |
| Pika | 1.14 (2) | 1.12 (2) | 1.22 (2) | 0.92 (3) | 0.78 (3) | 1.06 (2) |
| Latte | 0.62 (5) | 0.89 (4) | 0.84 (5) | 1.30 (2) | 1.15 (2) | 0.82 (3) |
| TF-T2V | 0.83 (4) | 0.80 (5) | 0.89 (4) | 0.72 (4) | 0.46 (4) | 0.65 (4) |
| Videocrafter2 | 0.87 (3) | 0.96 (3) | 1.01 (3) | 0.54 (5) | 0.29 (5) | 0.65 (5) |

Table 13: Prompt Category - Plant

| Model | Video Quality | Temporal Quality | Motion Quality | Text Alignment | Ethical Robustness | Human Preference |
|---|---|---|---|---|---|---|
| Gen2 | 2.07 (1) | 2.22 (1) | 2.11 (1) | 2.99 (1) | 4.01 (1) | 3.02 (1) |
| Pika | 1.05 (3) | 1.01 (3) | 1.03 (3) | 1.17 (3) | 0.85 (3) | 0.83 (3) |
| Latte | 1.04 (4) | 0.89 (4) | 1.07 (2) | 1.42 (2) | 1.67 (2) | 0.99 (2) |
| TF-T2V | 1.07 (2) | 1.04 (2) | 0.87 (4) | 0.78 (4) | 0.59 (4) | 0.70 (4) |
| Videocrafter2 | 0.71 (5) | 0.73 (5) | 0.78 (5) | 0.41 (5) | 0.22 (5) | 0.44 (5) |

Table 14: Prompt Category - Scenery

| Model | Video Quality | Temporal Quality | Motion Quality | Text Alignment | Ethical Robustness | Human Preference |
|---|---|---|---|---|---|---|
| Gen2 | 2.30 (1) | 2.37 (1) | 2.21 (1) | 3.31 (1) | 3.66 (1) | 2.56 (1) |
| Pika | 0.88 (3) | 0.97 (2) | 1.01 (2) | 0.78 (3) | 0.94 (3) | 0.98 (3) |
| Latte | 0.88 (4) | 0.91 (3) | 0.81 (5) | 1.04 (2) | 1.37 (2) | 1.02 (2) |
| TF-T2V | 1.08 (2) | 0.83 (4) | 0.95 (3) | 0.73 (4) | 0.50 (4) | 0.92 (4) |
| Videocrafter2 | 0.73 (5) | 0.74 (5) | 0.85 (4) | 0.38 (5) | 0.30 (5) | 0.56 (5) |

Table 15: Prompt Category - Vehicles

| Model | Video Quality | Temporal Quality | Motion Quality | Text Alignment | Ethical Robustness | Human Preference |
|---|---|---|---|---|---|---|
| Gen2 | 2.37 (1) | 2.49 (1) | 2.50 (1) | 2.62 (1) | 3.50 (1) | 2.97 (1) |
| Pika | 1.24 (2) | 1.32 (2) | 1.23 (2) | 0.98 (3) | 0.73 (3) | 0.96 (2) |
| Latte | 0.73 (5) | 0.70 (5) | 0.86 (3) | 1.62 (2) | 1.09 (2) | 0.72 (3) |
| TF-T2V | 0.77 (4) | 0.74 (3) | 0.67 (5) | 0.48 (5) | 0.47 (4) | 0.54 (4) |
| Videocrafter2 | 0.93 (3) | 0.74 (4) | 0.77 (4) | 0.62 (4) | 0.30 (5) | 0.49 (5) |

