# OpenReview forum: "Rethinking Human Evaluation Protocol for Text-to-Video Models: Enhancing Reliability, Reproducibility, and Practicality"
_NeurIPS.cc/2024/Conference — NeurIPS 2024 poster_

### Official Review · Reviewer_LQLU · 2024-06-22

**Soundness:** 4
**Presentation:** 4
**Contribution:** 4
**Rating:** 8
**Confidence:** 2

**Summary:**

This paper introduces the Text-to-Video Human Evaluation (T2VHE) protocol, a standardized approach for assessing the quality of videos generated from text.
Besides, this paper comprehensively evaluates some famous text-to-video models with their proposed reliable human assessment, which is very exciting and valuable for this community.

**Strengths:**

1. Standardized human evaluation protocol: T2VHE provides a comprehensive and consistent framework for evaluating T2V models, enabling fair and reliable comparisons between different models.
2. Reduced annotation costs: The dynamic evaluation module reduces the number of annotations needed by approximately 50%, making it more practical for researchers to evaluate large numbers of models.
3. Comprehensive evaluation on existing text-to-video models, revealing some interesting findings.

**Weaknesses:**

1. The algorithm’s performance and stability may vary depending on the specific prompt types and model characteristics.
2. The ELO rating [1] mechanism seems to be a competitive baseline for the Rao and Kupper method used in this paper, which has been widely used in the comparsion of LLMs. Could you please describe the difference between Rao and Kupper method and ELO rating, and predict which kinds of method is better?

[1] Chatbot Arena: Benchmarking LLMs in the Wild with Elo Ratings (https://lmsys.org/blog/2023-05-03-arena)

**Questions:**

1. How does the performance of the evaluated models vary across different domains or types of prompts?

---

> ### Author Rebuttal · Authors · 2024-08-07
>
> We sincerely thank reviewer LQLU for the valuable comments and give detailed responses to all questions. Due to space constraints, we submit part of the replies in "Official Comment", for convenience, we have summarized conclusions for each reply. Looking forward to further discussion with you.
>
> **W1: The algorithm’s performance and stability may vary depending on the specific prompt types and model characteristics.**
>
> Thank you for your insightful feedback, which will undoubtedly contribute to the improvement and clarity of our work. For the concerns and questions, here are our responses:
>
> **Analysis:**
>
> - Since our dynamic algorithm is designed for human evaluation protocols, as with all assessment protocols, it is inevitable that for different assessment contents, i.e., using different prompts and different models, there will be an impact on the final assessment results, but **this impact is limited**.
>
> - To reduce the impact of different types of prompts on algorithm performance, we utilize the Prompt Suite per Category [4] as a source of prompts, which comprises prompts manually curated from eight distinct categories. This approach ensures that the performance of the algorithm will not be affected too much as long as the source of prompts is comprehensive when using this dynamic evaluation component.
>
> - As for the effect of different model performances on the algorithm's performance, as shown in Table 3, the number of comparison pairs discarded will be smaller if a set of models with similar performances is used for the evaluation, while the algorithm's effect will be more significant if the performance gap between the models to be compared is obvious. This phenomenon is in line with our starting point, i.e., no more cost is wasted on evaluating combinations with significant differences in strength, whereas more comparisons are needed to obtain accurate results for combinations with close performance.
>
> - Furthermore, we employ bootstrap confidence intervals for the score estimates of each model across various metrics. Section 5 and Figure 3 of our paper demonstrate that **even in extreme cases, our rank estimation is consistent and trustworthy**.
>
>
>
> **Conclusion:**
>
> - The effect of different prompts and model characteristics on the algorithm is limited, and experimental results demonstrate that **this effect does not diminish the validity and reliability** of the dynamic evaluation component at all.
>
>
>
> **W2: Could you please describe the difference between Rao and Kupper method and ELO rating, and predict which kind of method is better?**
>
> Thank you for highlighting the comparison between the ELO rating mechanism and the Rao and Kupper model. We appreciate the opportunity to clarify the differences and discuss the suitability of each method for our specific application.
>
> **Analysis:**
>
> - **Comparison Between ELO Rating and Rao & Kupper Model.** The ELO rating system is a widely recognized method for ranking and scoring based on pairwise comparisons, especially in competitive environments such as chess and, more recently, in comparing large language models (LLMs) [1]. However, the Rao and Kupper model [2], which we have chosen for this study, is **a modification of the Bradley-Terry model**[3] that accommodates tied comparison results, making it **more flexible** in scenarios where outcomes are not always decisive.
>
> - **Handling Tied Comparisons:**
> 	- **ELO Rating:** Typically, the ELO system is designed for scenarios where a clear winner and loser are defined. It does not inherently handle ties as effectively as the Rao and Kupper model.
> 	- **Rao & Kupper Model:** This model explicitly accounts for tied results, which is particularly beneficial in our context where annotators may find two models equally effective. This capability to handle ties allows for more accurate and nuanced model rankings based on the available data.
>
> - **Statistical Robustness:**
> 	- **ELO Rating:** The ELO system assumes a relatively straightforward model of win/loss, which can lead to biases when dealing with incomplete data or when the matches are not representative of all possible pairings.
> 	- **Rao & Kupper Model:** Our approach using the Rao and Kupper model is statistically robust against incomplete comparison data. By leveraging the maximum likelihood estimation process, it provides a more reliable ranking even with missing or tied data.
>
> - **Dynamic Pair Selection.** Both models allow for the dynamic selection of model pairs. However, the ability of the Rao and Kupper model to account for ties gives it an edge in scenarios where comparisons may not always yield a clear winner.
>
> - **Predicting Method Suitability.** Given the nature of our task, which involves nuanced evaluations where ties are possible and even expected, **the Rao and Kupper model is more suited to our needs**. While the ELO rating system is competitive and has been successfully applied in many domains, the ability of the Rao and Kupper model to handle ties and its robustness against incomplete data makes it more appropriate for our use case.
>
>
>
> **Conclusion:**
>
> - We hope this detailed explanation clarifies our methodology and addresses any concerns about the application of the Rao and Kupper model in our research. We also provide detailed parameter estimation details in Appendix C.3.

---

> ### Author Response · Authors · 2024-08-07
> **Rebuttal (2/3)**
>
> **Q1: How does the performance of the evaluated models vary across different domains or types of prompts?**
>
> Thanks for the comments.
>
> We conducted experiments on the performance of the evaluated models that varied across different types of prompts. The results are shown in the tables below.
>
> **Setting:**
>
> - We input the human annotation results corresponding to the eight prompt categories into the Rao and Kupper model [2] to estimate and rank the model scores.
>
> **Results:**
>
> - We report the model scores and corresponding rankings obtained for the annotation results under the eight prompt categories.
>
> - Some of the scores in the table are the same and the result of different rankings is due to the retention of two decimal places in the report.
>
> **Analysis:**
>
> - Our results show that using different prompt sources does lead to slight differences in evaluation results, whereas models with a clear advantage, such as Gen2, are unaffected and have the highest ranking regardless of which prompt is used.
>
> - This result further highlights the importance of using class-wide prompts when using human evaluation protocols.
>
>
>
> **Conclusion:**
>
> - Our experimental results show that different prompt categories **do slightly affect the evaluation results** of model performance, but this effect is not obvious for models with high performance, such as Gen2.
>
>
>
> **References:**
>
> [1] Chatbot Arena.
>
> [2] Ties in paired-comparison experiments: A generalization of the Bradley-Terry model. _Journal of the American Statistical Association_, _62_(317), 194-204.
>
> [3] Rank analysis of incomplete block designs: I. The method of paired comparisons. _Biometrika_, _39_(3/4), 324-345.
>
> [4] Vbench: Comprehensive benchmark suite for video generative models.  CVPR 2024.
>
>
>
>
>
> Prompt Category - Animal
>
> | |Video Quality|Temporal Quality|Motion Quality|Text Alignment|Ethical Roubustness|Human Preference|
> |-|-|-|-|-|-|-|
> | Gen2          | 2.02(1)       | 2.15(1)          | 1.89(1)        | 2.40(1)        | 2.48(1)             | 3.38(1)          |
> | Pika          | 1.18(2)       | 1.41(2)          | 1.52(2)        | 1.54(2)        | 1.04(2)             | 1.35(2)          |
> | Latte         | 0.90(4)       | 0.98(3)          | 0.92(4)        | 1.29(3)        | 0.92(3)             | 1.01(3)          |
> | TF-T2V        | 0.98(3)       | 0.93(4)          | 0.96(3)        | 0.99(4)        | 0.73(4)             | 0.52(4)          |
> | Videocrafter2 | 0.84(5)       | 0.72(5)          | 0.86(5)        | 0.53(5)        | 0.69(5)             | 0.35(5)          |
>
>
> Prompt Category - Architecture
>
> | |Video Quality|Temporal Quality|Motion Quality|Text Alignment|Ethical Roubustness|Human Preference|
> |-|-|-|-|-|-|-|
> | Gen2          | 2.12(1)       | 1.86(1)          | 1.95(1)        | 2.49(1)        | 3.45(1)             | 2.54(1)          |
> | Pika          | 1.00(3)       | 1.46(2)          | 1.13(2)        | 0.92(3)        | 0.59(3)             | 1.05(2)          |
> | Latte         | 0.81(5)       | 1.07(4)          | 0.93(4)        | 1.28(2)        | 1.30(2)             | 0.95(3)          |
> | TF-T2V        | 0.82(4)       | 1.10(3)          | 0.94(3)        | 0.85(4)        | 0.50(4)             | 0.72(4)          |
> | Videocrafter2 | 1.09(2)       | 0.73(5)          | 0.89(5)        | 0.62(5)        | 0.25(5)             | 0.62(5)          |
>
>
> Prompt Category - Food
>
> | |Video Quality|Temporal Quality|Motion Quality|Text Alignment|Ethical Roubustness|Human Preference|
> |-|-|-|-|-|-|-|
> | Gen2          | 2.34(1)       | 2.39(1)          | 2.00(1)        | 2.87(1)        | 3.45(1)             | 3.04(1)          |
> | Pika          | 0.99(2)       | 1.03(2)          | 1.16(2)        | 0.94(3)        | 0.59(3)             | 0.82(3)          |
> | Latte         | 0.89(3)       | 0.92(3)          | 0.83(5)        | 1.57(2)        | 1.30(2)             | 1.42(2)          |
> | TF-T2V        | 0.79(5)       | 0.74(4)          | 1.04(3)        | 0.51(4)        | 0.50(4)             | 0.47(4)          |
> | Videocrafter2 | 0.80(4)       | 0.74(5)          | 0.87(4)        | 0.47(5)        | 0.25(5)             | 0.35(5)          |

---

> ### Author Response · Authors · 2024-08-07
> **Rebuttal (3/3)**
>
> Prompt Category - Human
>
> | |Video Quality|Temporal Quality|Motion Quality|Text Alignment|Ethical Roubustness|Human Preference|
> |-|-|-|-|-|-|-|
> | Gen2          | 2.44(1)       | 2.64(1)          | 2.14(1)        | 2.96(1)        | 3.66(1)             | 2.98(1)          |
> | Pika          | 1.38(2)       | 1.29(2)          | 1.52(2)        | 0.99(3)        | 0.78(3)             | 1.23(2)          |
> | Latte         | 0.59(5)       | 0.62(5)          | 0.75(5)        | 1.38(2)        | 0.96(2)             | 0.44(5)          |
> | TF-T2V        | 0.94(4)       | 0.79(3)          | 0.91(3)        | 0.61(4)        | 0.37(4)             | 0.62(3)          |
> | Videocrafter2 | 0.97(3)       | 0.64(4)          | 0.87(4)        | 0.45(5)        | 0.27(5)             | 0.55(4)          |
>
>
> Prompt Category - Lifestyle
>
> | |Video Quality|Temporal Quality|Motion Quality|Text Alignment|Ethical Roubustness|Human Preference|
> |-|-|-|-|-|-|-|
> | Gen2          | 2.52(1)       | 2.11(1)          | 1.99(1)        | 2.69(1)        | 3.59(1)             | 2.63(1)          |
> | Pika          | 1.14(2)       | 1.12(2)          | 1.22(2)        | 0.92(3)        | 0.78(3)             | 1.06(2)          |
> | Latte         | 0.62(5)       | 0.89(4)          | 0.84(5)        | 1.30(2)        | 1.15(2)             | 0.82(3)          |
> | TF-T2V        | 0.83(4)       | 0.80(5)          | 0.89(4)        | 0.72(4)        | 0.46(4)             | 0.65(4)          |
> | Videocrafter2 | 0.87(3)       | 0.96(3)          | 1.01(3)        | 0.54(5)        | 0.29(5)             | 0.65(5)          |
>
>
> Prompt Category - Plant
>
> | |Video Quality|Temporal Quality|Motion Quality|Text Alignment|Ethical Roubustness|Human Preference|
> |-|-|-|-|-|-|-|
> | Gen2          | 2.07(1)       | 2.22(1)          | 2.11(1)        | 2.99(1)        | 4.01(1)             | 3.02(1)          |
> | Pika          | 1.05(3)       | 1.01(3)          | 1.03(3)        | 1.17(3)        | 0.85(3)             | 0.83(3)          |
> | Latte         | 1.04(4)       | 0.89(4)          | 1.07(2)        | 1.42(2)        | 1.67(2)             | 0.99(2)          |
> | TF-T2V        | 1.07(2)       | 1.04(2)          | 0.87(4)        | 0.78(4)        | 0.59(4)             | 0.70(4)          |
> | Videocrafter2 | 0.71(5)       | 0.73(5)          | 0.78(5)        | 0.41(5)        | 0.22(5)             | 0.44(5)          |
>
>
> Prompt Category - Scenery
>
> | |Video Quality|Temporal Quality|Motion Quality|Text Alignment|Ethical Roubustness|Human Preference|
> |-|-|-|-|-|-|-|
> | Gen2          | 2.30(1)       | 2.37(1)          | 2.21(1)        | 3.31(1)        | 3.66(1)             | 2.56(1)          |
> | Pika          | 0.88(3)       | 0.97(2)          | 1.01(2)        | 0.78(3)        | 0.94(3)             | 0.98(3)          |
> | Latte         | 0.88(4)       | 0.91(3)          | 0.81(5)        | 1.04(2)        | 1.37(2)             | 1.02(2)          |
> | TF-T2V        | 1.08(2)       | 0.83(4)          | 0.95(3)        | 0.73(4)        | 0.50(4)             | 0.92(4)          |
> | Videocrafter2 | 0.73(5)       | 0.74(5)          | 0.85(4)        | 0.38(5)        | 0.30(5)             | 0.56(5)          |
>
>
> Prompt Category - Vehicles
>
> | |Video Quality|Temporal Quality|Motion Quality|Text Alignment|Ethical Roubustness|Human Preference|
> |-|-|-|-|-|-|-|
> | Gen2          | 2.37(1)       | 2.49(1)          | 2.50(1)        | 2.62(1)        | 3.50(1)             | 2.97(1)          |
> | Pika          | 1.24(2)       | 1.32(2)          | 1.23(2)        | 0.98(3)        | 0.73(3)             | 0.96(2)          |
> | Latte         | 0.73(5)       | 0.70(5)          | 0.86(3)        | 1.62(2)        | 1.09(2)             | 0.72(3)          |
> | TF-T2V        | 0.77(4)       | 0.74(3)          | 0.67(5)        | 0.48(5)        | 0.47(4)             | 0.54(4)          |
> | Videocrafter2 | 0.93(3)       | 0.74(4)          | 0.77(4)        | 0.62(4)        | 0.30(5)             | 0.49(5)          |

---

> ### Author Response · Authors · 2024-08-11
> **To save reviewer's time, we put a summary of rebuttal**
>
> Dear Reviewer LQLU,
>
> Thanks so much again for the time and effort in our work. Considering the limited time available and to save the reviewer's time, we **summarized our responses here**.
>
> 1. \[**The algorithm’s performance may vary depending on the specific prompt types and model characteristics** \]:
>
>    **Response:**
>    - The effect of different prompts and model characteristics on the algorithm is limited, and experimental results demonstrate that this effect does not diminish the validity and reliability of the dynamic evaluation component at all.
>
> 2. \[**Compare Rao and Kupper method and ELO rating** \]:
>
>    **Response:**
>    - While the ELO rating system is competitive and has been successfully applied in many domains, given the nature of our task, which involves nuanced evaluations where ties are possible and even expected, the Rao and Kupper model is more suited to our needs.
>
> 3. \[**How does the performance of the evaluated models vary across different domains or types of prompts** \]:
>
>    **Response:**
>    - We report the model scores and corresponding rankings obtained for the annotation results under the eight prompt categories.
>    - The results show that different prompt categories do slightly affect the evaluation results of model performance, but this effect is not obvious for models with high performance, such as Gen2.
>
>
> Since the discussion stage is already halfway through, may I know if our rebuttal addresses the concerns? If there are further concerns or questions, we are more than happy to address them. Thanks again for taking the time to review our work and provide insightful comments.
>
> Best Regards,
>
> Authors

---

> ### Comment · Reviewer_LQLU · 2024-08-12
> **Response to Authors**
>
> Thank you for the detailed responses from authors. I have acknowledged the strengths of your wonderful paper and I will keep my score.

---

> > ### Author Response · Authors · 2024-08-12
> > **We appreciate the support!**
> >
> > Thanks very much for **supporting and accepting** our work. We are happy that we have addressed your concern. Your constructive feedback is very helpful for us in improving our work.
> >
> > If there are any further questions, we are more than willing to provide a detailed explanation. Thank you!

---

### Official Review · Reviewer_RMCR · 2024-07-10

**Soundness:** 3
**Presentation:** 3
**Contribution:** 2
**Rating:** 5
**Confidence:** 5

**Summary:**

This manuscript presents a novel, standardized human evaluation protocol specifically tailored for Text-to-Video (T2V) models. The protocol encompasses a meticulously designed suite of evaluation metrics supplemented with robust annotator training resources. The effective deployment of Latent Action Ratings (LARs) is a notable feature of this work, enabling the acquisition of high-quality annotations. A dynamic evaluation component is also introduced, which remarkably curtails annotation costs by approximately 50% while maintaining the integrity of annotation quality. The paper further offers an exhaustive evaluation of cutting-edge T2V models. A commitment to open-source the entire evaluation process and associated code is a commendable initiative, which will undoubtedly empower the broader research community to assess novel models using updated data, building upon existing reviews.

**Strengths:**

+ First of all, I am very grateful to the authors for their great contribution to the open-source community. I believe that this work will help existing research take a bigger step after it is released;
+ Then, this paper carefully considers the evaluation criteria of T2V. 4 objective and 2 subjective annotation evaluations are used to improve the quality of discrimination, making the annotation results more in line with human preferences and lower error rates or bias;
+ Furthermore, this work maximizes the effectiveness of comparative evaluation. From my point of view, this is very much like doing a reward model that serves for reinforcement learning from human feedback (RLHF), which will help unify and simplify the difficulty of annotation and quantification;
+ Finally, due to a large amount of human power investment, the authors introduced a dynamic evaluation component to reduce the annotation cost and claimed that the expense could be reduced by 50% at the same quality.

**Weaknesses:**

— My biggest concern is that the evaluation metrics currently proposed by the authors cannot be proven reasonable or convincing. Although this manuscript has used several quality standards (4 objective and 2 subjective), are they complete or redundant? For example, I think "Video Quality" is a vast range, which may include multiple aspects such as aesthetics, clarity, and even another indicator, "Text Alignment" (because if the generated elements do not match the prompt, it will not be considered a good video); I think "Temporal Quality" and "Motion Quality" are not two completely independent metrics, they should be highly related and coupled.
— Furthermore, the authors mentioned that this paper adopts the Rao and Kupper model to quantify annotations results. I tried to follow this paper to understand how to quantify these metrics whose weights I am not sure about. Still, unfortunately, I found no similarities between it and this paper. I hope the authors can scientifically and clearly explain how the metrics are involved in the final results, which is one of the core contributions of this paper.
— In addition, since the annotated results of all metrics are labeled by manual A/B testing, is the human preference the only one that dominates? How can the aesthetics and evaluation systems of different annotators be consistent? The same problem arises in the reward model, where its approach has been verified in the LLM field and cannot achieve high accuracy (only about 70+%). Thus, it needs to be used in conjunction with PPO/DPO. As this manuscript constructs datasets this way,  how can we avoid the problem?
— Finally, although the dynamic evaluation component proposed in this article reduces annotation costs to approximately half,  the annotation task is too dependent on manual work; I think its promotion and ease of use in the open-source community are still questionable.

**Questions:**

I acknowledge that I appreciate the authors' contributions to the open source community, and reasonable and practical answers to the concerns in the "Weaknesses" section will help improve my rating.

**Limitations:**

This manuscript has discussed the relevant limitations, please refer to the "Weaknesses" section for more details.

---

> ### Author Rebuttal · Authors · 2024-08-07
>
> We sincerely thank reviewer RMCR for the valuable comments and give detailed responses to all questions. Due to space constraints, we submit part of the replies in "Official Comment", for convenience, we have summarized conclusions for each reply. Looking forward to further discussion with you.
>
> **W1: The evaluation metrics currently proposed by the authors cannot be proven reasonable or convincing.**
>
> Thanks for the comments. To build a reasonable evaluation system, we have investigated over 200 papers and found that due to the inconsistency of definition, the evaluation systems differ greatly. For example, for the same metric "Video Quality", some split it into eight sub-metrics, while some define it as an overall preference, as in the example given by the reviewer. Therefore, to **balance the practicality and comprehensiveness** of the protocol, we analyze this from the following perspectives.
>
> **Analysis:**
>
> - **Decomposition of "Video Quality".** For the T2V models, one of the most basic requirements is that the generated content should be consistent with the prompt, i.e., "Text Alignment", and the second one is logical, including whether the action conforms to physical laws and whether it has a certain degree of continuity, i.e., "Motion Quality" and "Temporal Quality". Based on these, the creators usually want the video content to have certain characteristics, i.e., "Video Quality" in the narrow sense.
> - **Division of "Motion Quality" and "Temporal Quality".** We observe that many studies are focusing on how to keep the objects continuous and flicker-free between frames. Measuring this ability separately from motion quality can help to compare and analyze these models more fairly, e.g., placing studies focusing on long video generation [1] under a single metric with studies focusing on motion control [2] is unfair to both studies.
> - To further improve the definition of the four objective metrics, we investigated many benchmark studies [3-6], which were **also carried out from these four aspects**, and further developed the corresponding reference angles. However, in further research into the relatively mature fields [7,9], we find that existing T2V evaluation systems lack the design of these subjective metrics, i.e., "Ethical Robustness" and "Human Preference".
> - **Necessity of subjective metrics.** Due to the diversity of users, ethical review of the generated content is an important part of T2I model evaluation [7]. Unfortunately, our investigation shows all current T2V evaluation protocols missed this consideration. Therefore, we refer to the research ideas of T2I and formulate corresponding definitions and reference angles to make up for this lack. Similarly, human preferences should be assessed independently of objective metrics because of differences in individual preferences, e.g., even if some videos are of poor quality, the interesting content can still give the model practical value, which should not be overlooked during the evaluation.
> - **Metric granularity.** To train a reliable scorer, VBench [4] further divides its objective metric, producing 16 sub-metrics. However, this practice **does not apply to the design of human evaluation protocols**, as annotators usually have strong abstract capabilities, just proper training and guidance can help them fully understand the indicators and produce reliable annotations [9]. Therefore, to reduce the annotation cost and improve the practicality of our protocol, we did not design more disaggregated metrics.
>
> **Conclusion:**
>
> - Through extensive research and rigorous analysis, we designed an assessment system that not only **incorporates the mainstream metrics** used in existing T2V studies but also **complements the missing parts** of previous studies, making our evaluation metrics reasonable or convincing.

---

> ### Author Response · Authors · 2024-08-07
> **Rebuttal (2/3)**
>
> **W2: Explain how Rao and Kupper model quantifies annotations results.**
>
> We appreciate your feedback on our description of the Rao and Kupper model [10]. We understand the importance of clarity and detail in explaining our methods. Here’s a more comprehensive overview of the model.
>
> **Analysis:**
>
> - **Probabilistic Model Overview:**
>
>   - **Model Definition:** The Rao and Kupper model is a probabilistic framework used for pairwise comparisons, allowing us to estimate scores for each model and rank them accordingly.
>   - **Maximum Likelihood Estimation:** We employ maximum likelihood estimation to derive the scores, which provides a statistical basis for the rankings.
> - **Detailed Explanation:**
>
>   - **Comparison Process:** Consider a scenario with $t$ models. For any pair of models $i$ and $j$, the notation $i \succ j$ indicates $i$ beating $j$, $i \prec j$ indicates $i$ losing to $j$, and $i \sim j$ indicates a tie. The probabilities of these events are defined as follows:
>     $$
>      \begin{align}
>      P(i \succ j) &= \frac{p_i}{p_i + \theta p_j},
>      P(i \sim j) &= \frac{p_i p_j (\theta^2 -1)}{(p_i + \theta p_j)(\theta p_i + p_j)},
>      \end{align}
>     $$
>      where $p_i$ and $p_j$ represent the positive real-valued scores of models $i$ and $j$. $\theta$ is a parameter indicating the likelihood of a tie between closely matched models.
>
>    - **Estimation Method:** The likelihood function for our model is constructed as:
>      $$
>      \begin{equation}
>      L(p, \theta) = \prod_{i = 1}^{t} \prod_{j = i + 1}^{t} \left( \frac{p_i}{p_i + \theta p_j} \right)^{n_{ij}} \left( \frac{p_j}{\theta p_i + p_j} \right)^{n_{ji}} \left( \frac{p_i p_j (\theta^2 -1)}{(p_i + \theta p_j)(\theta p_i + p_j)} \right)^{\tilde{n}_{ij}},
>      \end{equation}
>      $$
>
>      and the corresponding log-likelihood function is:
>
>      $$
>      \begin{equation}
>      l(p, \theta) = \sum_{i = 1}^{t} \sum_{j = i + 1}^{t} \left( n_{ij} \log \frac{p_i}{p_i + \theta p_j} + n_{ji} \log \frac{p_j}{\theta p_i + p_j} + \tilde{n}_{ij} \log \frac{p_i p_j (\theta^2 -1)}{(p_i + \theta p_j)(\theta p_i + p_j)} \right),
>      \end{equation}
>      $$
>      where $p \in \mathbb{R}^t$ is a vector of scores $(p_1, \dots, p_t)^T$. We estimate $p$ and $\theta$ by maximizing the log-likelihood function numerically.
>
> - **Application and Results:**
> 	- **Model Ranking:** After estimating the scores, models are ranked based on these scores, reflecting their relative strengths. Our detailed explanation in Appendix C.3 covers the model fitting process and the derivation of scores and rankings.
>
>
> **Conclusion:**
>
> - We hope this detailed explanation clarifies our methodology and addresses any concerns about the application of the Rao and Kupper model in our research. We also provide detailed parameter estimation details in Appendix C.3.

---

> ### Author Response · Authors · 2024-08-07
> **Rebuttal (3/3)**
>
> **W3:  Is the human preference the only one that dominates? How can the aesthetics and evaluation systems of different annotators be consistent? How can we avoid the problem?**
>
> Thanks for the comments. We analyze this from the following perspectives.
>
> **Analysis:**
>
> - We observed similar problems in human evaluation studies in the T2I and NLG communities [8,9,12], as shown in the footnotes of the article, some NLG studies [11] achieve an IAA of only 0.14, i.e., even if constructing sufficiently objective protocols and using the most professional annotators, one **cannot avoid** the biases brought by human preferences.
>
> - At the same time, however, it has also been shown that **the existence of human preferences is a vital part of human assessment** [9], i.e., a good protocol should both effectively reduce bias due to preferences and make reservations about this, as too much agreement (e.g., IAA greater than 0.7) may suggest that there is a lack of diversity among annotators.
>
> - This is one of the core reasons our protocol distinguishes between subjective and objective metrics. For **objective** metrics, we minimize the influence of personal preferences through detailed metric definitions and annotator training, whereas for **subjective** metrics, we encourage annotators to evaluate based on their preferences. Unlike objective metrics, subjective metrics are difficult to be replaced by automated metrics and reflect the diversity of annotators.
>
> **Conclusion:**
>
> - Previous studies and our experimental results have shown that **human preferences cannot and need not be eliminated** during evaluation. By providing different guidelines for different types of metrics, our protocol achieves a balance between the pros and cons of human preferences.
>
> **W4: The annotation task with the dynamic evaluation component is too dependent on manual work, its promotion and ease of use in the open-source community are still questionable.**
>
> Thanks for the comments. We analyze this from the following perspectives.
>
> **Analysis:**
>
> - Our dynamic evaluation component was proposed to improve human evaluation protocols, i.e., this method should be compared to a traditional human evaluation protocol that **also relies on manual work**, and uses **all the video pairs**. Our component can reduce the annotation cost without additional intervention during the annotation phase, which is important for popularizing human assessment.
>
> - For example, a researcher needs to recruit five annotators, each of whom can only evaluate 200 sets of video pairs; using the same budget, employing our dynamic component would allow the researcher to recruit 10 or more people to execute evaluation, as the workload for each individual is only about half of what it would otherwise be. Not only does this allow for a more varied and reliable set of annotated results, but also as the evaluation size increases, the effect of the component becomes more pronounced.
>
> - In addition, **this component is not limited to the T2V research field**, for the T2I and NLG research fields, by simply replacing the automated indicators in the pre-preparation phase, one can directly use this component in the customized protocol framework. We provide the detailed flow of the component operation in Appendix C.2 and will open-source all the related code. We also hope and encourage researchers from different fields to conduct more comprehensive human assessment protocols at a lower cost.
>
>
> **Conclusion:**
>
> - Our dynamic components are proposed to improve the efficiency of the human assessment process that **already requires manual work**, without additional expense. At the same time, it can be used in other research fields with **simple adjustments**.
>
>
>
> **References:**
>
> [1] Control-A-Video: Controllable Text-to-Video Generation with Diffusion Models.
>
> [2] ControlVideo: Training-free Controllable Text-to-Video Generation. ICLR 2024.
>
> [3] EvalCrafter: Benchmarking and Evaluating Large Video Generation Models. CVPR 2024.
>
> [4] VBench: Comprehensive Benchmark Suite for Video Generative Models. CVPR 2024.
>
> [5] AIGCBench: Comprehensive Evaluation of Image-to-Video Content Generated by AI. TBench 2024.
>
> [6] FETV: A Benchmark for Fine-Grained Evaluation of Open-Domain Text-to-Video Generation. NIPS 2023.
>
> [7] Holistic Evaluation of Text-to-Image Models. NIPS 2023.
>
> [8] Toward Verifiable and Reproducible Human Evaluation for Text-to-Image Generation. CVPR 2023.
> [9] Evaluation of Text Generation: A Survey.
>
> [10] Ties in paired-comparison experiments: A generalization of the Bradley-Terry model. _Journal of the American Statistical Association_, _62_(317), 194-204.
>
> [11] The Perils of Using Mechanical Turk to Evaluate Open-Ended Text Generation. EMNLP 2021.
>
> [12] All That's 'Human' Is Not Gold: Evaluating Human Evaluation of Generated Text. IJCNLP 2021.

---

> ### Author Response · Authors · 2024-08-11
> **To save reviewer's time, we put a summary of rebuttal**
>
> Dear Reviewer RMCR,
>
> Thanks so much again for the time and effort in our work. Considering the limited time available and to save the reviewer's time, we **summarized our responses here**.
>
> 1. \[**The evaluation metrics cannot be proven reasonable or convincing** \]:
>
>    **Response:**
>    - Through extensive research and rigorous analysis, we designed an assessment system that not only incorporates the mainstream metrics used in existing T2V studies but also complements the missing parts of previous studies, making our evaluation metrics reasonable and convincing.
>
> 2. \[**Explain how Rao and Kupper model quantifies annotations results** \]:
>
>    **Response:**
>    - We provide detailed explanation in Rebuttal (2/3) and detailed parameter estimation details in Appendix C.3.
>
> 3. \[**How to handle human preference** \]:
>
>    **Response:**
>    - Previous studies and our experimental results have shown that human preferences cannot and need not be eliminated during evaluation.
>    - By providing different guidelines for different types of metrics, our protocol achieves a balance between the pros and cons of human preferences.
>
> 4. \[**Dynamic evaluation component is too dependent on manual work** \]:
>
>    **Response:**
>    - Our dynamic components are proposed to improve the efficiency of the human assessment process that already requires manual work, without additional expense. At the same time, it can be used in other research fields with simple adjustments.
>
>
> Since the discussion stage is already halfway through, may I know if our rebuttal addresses the concerns? If there are further concerns or questions, we are more than happy to address them. Thanks again for taking the time to review our work and provide insightful comments.
>
> Best Regards,
>
> Authors

---

> ### Author Response · Authors · 2024-08-12
> **Looking forward to the reply**
>
> Dear Reviewer RMCR,
>
> As we draw closer to the rebuttal deadline, we would like to inquire if there are any additional questions or concerns about our work. We greatly value the feedback.
>
> Thanks again for the time and effort in our work!
>
> Best wishes,
>
> Authors

---

> > ### Comment · Reviewer_RMCR · 2024-08-13
> >
> > Thanks. The authors have addressed most of my concerns, so I'll raise my rating.

---

> > > ### Author Response · Authors · 2024-08-13
> > > **Thanks for your support!**
> > >
> > > Thank you very much for your **support and for accepting** our work. We are pleased to hear that we have successfully addressed your concerns. Your constructive feedback has been invaluable in helping us improve our research.
> > >
> > > If there are any further questions or if you require a more detailed explanation of any aspect of our work, we are more than willing to provide any additional information you may need.

---

### Official Review · Reviewer_AEKZ · 2024-07-11

**Soundness:** 3
**Presentation:** 3
**Contribution:** 2
**Rating:** 5
**Confidence:** 4

**Summary:**

This paper introduces the Text-to-Video Human Evaluation (T2VHE) protocol, a standardized approach for evaluating text-to-video (T2V) models, addressing the challenges posed by recent advancements in T2V technology. The T2VHE protocol supposedly offers a solution through well-defined metrics, annotator training, and an effective dynamic evaluation module.

Experimental results suggest that this protocol can offer high-quality annotations and also reduce evaluation costs. The authors plan to open-source the T2VHE protocol setup.

**Strengths:**

The paper addresses a timely issue in the rapidly evolving field of text-to-video (T2V) models: the lack of standardized quality assessment. Its strengths lie in providing a comprehensive literature review since 2016 and synthesizing past research into a unified evaluation protocol. This effort to aggregate previous work is particularly valuable, given the growing importance of T2V models. By proposing a standardized approach, the authors contribute to filling a critical gap in the field, potentially accelerating progress in T2V model development and evaluation. The highlighted strengths are as follows:

* It introduces a novel, comprehensive protocol (T2VHE) for human evaluation of text-to-video (T2V) models, addressing a significant gap in the field.
* The paper combines existing ideas (e.g., comparative scoring, Rao and Kupper model) in creative ways to improve the evaluation process.
* The authors conducted a thorough survey of 89 papers on video generation modeling, providing a solid foundation for their work.:
* The paper is generally well-structured and clearly written:
* The introduction provides a clear motivation for the work and outlines the main contributions.
* The authors commit to open-sourcing the entire evaluation process and code, which is crucial for reproducibility and further advancement of the field.

**Weaknesses:**

Major Weaknesses
* Lack of comparison with existing evaluation protocols: While the paper introduces a new protocol, it doesn't directly compare its performance or efficiency against existing evaluation methods. This makes it difficult to quantify the improvements over current practices. In addition, it lacks an understanding with evaluation in other domains. ex. T2I, NLP, or CV. With the lack of details or standards in T2V evaluation, it would have benefitted from an expansive understanding of human annotation as a whole. Thus, the ideas although well representative of the current state of evaluation in T2V, lacks the maturity, which affords the novelty or value, that hasn't been surveyed in other domains.
* Potential bias in dynamic evaluation: The dynamic evaluation module, while innovative, may introduce bias by discarding certain comparisons. As noted in Table 3, comparisons involving Gen2 were frequently omitted. This could potentially lead to less accurate estimations for top-performing models.
* Lack of error analysis: The paper doesn't provide a detailed analysis of cases where the protocol fails or produces inconsistent results. Understanding these edge cases could help improve the robustness of the evaluation method. This further dives into my understanding that although it is a great survey of existing protocols, lacks a contribution its aggregation. Further evaluation and consideration in other domains, would assure me that it is a standard that can be well accepted vs. one that is just more comprehensive than the past.

Minor Weakness
* In line 3, there's a small typo where "pop3 ularity" is split across lines. It should be "popularity".
* In line 31, "practicabil32 ity" is split across lines. This appears to be due to line wrapping rather than an actual error in the text.

**Questions:**

* Can you provide more details on how the dynamic evaluation module handles potential biases when discarding certain comparisons, especially for top-performing models like Gen2?
* How does the protocol ensure that the reduced number of annotations in the dynamic evaluation still captures a representative sample across different prompt categories and model capabilities?
* What steps were taken to validate that the training provided to LRAs effectively mitigates potential biases they may have, especially given their potential lack of diversity compared to crowdsourced annotators?
* How does the protocol account for potential cultural or linguistic biases in the evaluation of "Ethical Robustness"? Can you provide more details on how this metric is defined and measured?
* What considerations were made in designing the protocol to ensure its applicability across different types of T2V models (e.g., models specializing in different video styles or lengths)?
* Would the survey of non -T2V evaluation approaches be beneficial when merging with this technology? With the lack of maturity and rapidly evolving field, would not benefit from a cross-domain understanding from more mature fields?

**Limitations:**

Mostly adequately addressed limitations. Would like to see some understanding in regards to human annotation, than just T2V.

---

> ### Author Rebuttal · Authors · 2024-08-07
>
> We sincerely thank reviewer AEKZ for the valuable comments and give detailed responses to all questions. Due to space constraints, we submit part of the replies in "Official Comment", for convenience, we have summarized conclusions for each reply. Looking forward to further discussion with you.
>
> **W1:  Lack of comparison with existing evaluation protocols. Lacks an understanding of evaluation in other domains.**
>
> Thank you for your insightful comment. We analyze it from the following perspectives.
>
> **Analysis:**
>
> - **We compared our protocol with existing ones in lines 31-71 and Tables 6 and 7.** Specifically, existing human evaluation for video generation meets reproducibility, reliability, and practicability challenges. To address these issues, we designed a well-developed evaluation metrics system, provided detailed annotator training, introduced a novel dynamic evaluation component, and demonstrated the effectiveness of these designs through extensive and well-established annotation experiments. **As shown in Tables 1-3, our design not only effectively improves annotator consistency, but also reduces the annotation cost by about 50%.**
> - While we investigated many corresponding protocols in the T2I and NLG communities [1-3], we did not show them in detail in the paper due to space limits. These studies usually focus on the design of protocol metrics definitions, whereas our protocol not only builds a comprehensive evaluation system, but also explores the differences between professional annotators and lab recruiters, and how the differences can be bridged through effective annotator training. Furthermore, we propose a novel dynamic assessment component based on traditional protocols, which drastically reduces the cost of assessment. **These contributions have not been performed by other fields of assessment**.
> - Regarding the maturity of our protocol, we investigated more than 200 video generation literature, including but not limited to the fields of T2V, video editing, and frame prediction. After referring to these and the studies in the fields of T2I and NLG, we summarized a set of detailed evaluation metrics and training processes. Due to time and cost constraints, we did not extend the protocol to areas such as T2I and NLG, which will be our **future work**. We'll also add a discussion of this in the article.
>
> **Conclusion:**
>
> - Our protocol is based on a thorough investigation of human evaluation and video generation articles in different fields and is experimentally validated to have the advantages of **reproducibility, reliability, and practicability** compared to traditional protocols.
>
> **W2:  The dynamic evaluation module, may introduce bias by discarding certain comparisons.**
>
> Thank you for your insightful comment. We analyze it from the following perspectives.
>
> **Analysis:**
>
> - Our estimates of model scores are based on the Rao and Kupper model [4], which is **inherently capable of handling incomplete comparisons**. This property ensures that our ranking estimates remain reliable even when some comparisons are discarded.
> - We provide an in-depth analysis of this phenomenon in lines 325-340, the rigorous bootstrap confidence intervals indicate that **discarding comparisons with significant differences does not affect the estimation of the final ranking results**, as shown in Figure 3.
>
> **Conclusion:**
>
> - Theoretically, our protocol design allows for plausible estimation in the absence of numerous comparisons, and we further validate this capability through bootstrap confidence interval experiments.
>
> **W3: The paper doesn't provide a detailed analysis of cases where the protocol fails or produces inconsistent results. Need further evaluation and consideration in other domains.**
>
> Thank you for your insightful comment. We analyze it from the following perspectives.
>
> **Analysis:**
>
> - We analyze the situation when protocol produces inconsistent results in lines 244-253. Specifically, in cases where researchers use untrained lab recruiters, their annotation results may be inconsistent with those from crowdsourcing platforms and trained annotators, **highlighting the importance of the annotator training process we designed**.
> - We also provide an analysis of the dynamic component's reliability in lines 325-340, the rigorous bootstrap confidence intervals indicate that it can **guarantee the reliability** of the results while reducing the cost of annotation by about **50%**.
> - The proposed annotator training methods and dynamic components **are not limited to the T2V research field**, for the T2I and NLG research fields, by simply modifying related concepts and examples, one can directly use these in the customized protocol framework.
>
> **Conclusions:**
>
> - We analyzed the case of producing inconsistent results under the traditional protocol setup, demonstrating the importance of the annotator training we designed, and conducted a reliability analysis of the dynamic evaluation component, which proved that the introduction of this component does not result in inconsistent protocol results.
> - Our design is not only to provide a more reliable, reproducible, and practical T2V assessment protocol but also to provide ideas and reference materials for human evaluation research in **other fields**.
>
> **MW1&MW2: Small typo errors.**
>
> Thanks for the reminder. These typo errors are caused by automatic line breaks during latex compilation, not actual errors in the text editor.

---

> ### Author Response · Authors · 2024-08-07
> **Rebuttal (2/4)**
>
> **Q1: More details on how the dynamic evaluation module handles potential biases.**
>
> Thank you for your question regarding the dynamic evaluation module. We have designed our method with robustness and accuracy in mind. Here's how we address potential biases:
>
> **Analysis:**
>
> - **Robustness Against Incomplete Comparisons:**  It is well-documented that the Bradley-Terry model [5] is robust against incomplete comparison data. Our approach, based on the Rao and Kupper model [4], **inherits this robustness and is specifically designed to handle scenarios with missing or incomplete data**. This property ensures that our ranking estimates remain reliable even when some comparisons are discarded.
> - **Handling Top-Performing Models (e.g., Gen2):**
>   - **Bootstrap Confidence Intervals:** To mitigate potential biases in ranking models with incomplete data, we employ bootstrap confidence intervals for the score estimates of each model across various metrics. Section 5 and Figure 3 of our paper demonstrate that even in extreme cases, our rank estimation is consistent and trustworthy.
>   - **Ensuring Fairness:** By using bootstrap methods, we can quantify the uncertainty around our model rankings and ensure that our evaluation is fair and unbiased, even when certain comparisons are omitted.
> - **Transparency and Detail:**  We provide a detailed explanation of our dynamic evaluation module and its bias-handling mechanisms in the supplementary materials, ensuring that our methodology is transparent and replicable.
>
> **Conclusions:**
>
> - From a methodological design perspective, the Rao and Kupper model we used can **accurately estimate model scores using incomplete comparisons**.
> - From an experimental validation perspective, we conducted rigorous bootstrap confidence interval experiments, the results of which confirm that even after discarding many comparisons on top models such as Gen2, the final estimate of the relative strength of the model is **still accurate and credible**.
>
> **Q2: How to ensure that the reduced number of annotations still captures a representative sample?**
>
> Thank you for your insightful comment. We analyze it from the following perspectives.
>
> **Analysis:**
>
> - Before the dynamic evaluation begins, the protocol undergoes a static evaluation, in which the annotator prioritizes a group of samples that are worthy of being annotated. These video pairs contain the **full range of prompt categories and model combinations**, as shown in Figure 4, and have little difference in the scores of the automated indicators, i.e., they are **the most representative samples** according to the reviewers.
> - As these samples won't be discarded during the static evaluation, even if subsequent evaluations discard many comparisons, the final evaluation results can include all categories of prompts and the most representative comparisons.
>
> **Conclusion:**
>
> - The static evaluation in the protocol **retains the samples that are most worthy of being annotated**, ensuring that the evaluation results **won't be biased** by the lack of specific kinds of prompts and model combinations.
>
> **Q3: How to validate that the training provided to LRAs effectively mitigates potential biases and their potential lack of diversity?**
>
> Thank you for your insightful comment. We designed detailed guidelines and training processes to address this bias,  as following points:
>
> **Analysis:**
>
> - To help annotators understand the definition of each metric, we provide multiple **reference perspectives** for each indicator. For each reference perspective, we provide **detailed examples** including a set of reference videos, a description of the analysis process, and corresponding conclusions, so the annotators from different sources can evaluate with the same mode of thinking. The specific definition and training process are shown in Table 4, Figure 1 (c), and Figures 5 to 9, respectively.
> - As shown in Tables 1 and 2, the consistency between the trained annotators and the crowdsourced annotators is significantly improved, i.e., the metric examples and the training process can help eliminate the inconsistency caused by the potential bias.
> - Although the lack of LRA's diversity cannot be completely avoided [2], by distinguishing between subjective and objective indicators, we can **preserve the diversity of annotators** as much as possible during the evaluation process.
>
> **Conclusion:**
>
> - We minimize annotator bias in assessing objective metrics by providing **detailed guidelines and training processes**, and we retain annotator diversity by introducing **subjective metrics**.

---

> ### Author Response · Authors · 2024-08-07
> **Rebuttal (3/4)**
>
> **Q4: How does the protocol account for potential biases in the evaluation of "Ethical Robustness"? Can you provide more details on how this metric is defined and measured?**
>
> Thank you for your insightful comment. We analyze it from the following perspectives.
>
> **Analysis:**
>
> - Our survey shows that while the measurement of the ethics of generated content is an important evaluation metric for generative models [6], it is missing from existing T2V evaluation protocols. Therefore, after a thorough investigation of the relevant research [7,8], we defined the corresponding reference angles according to the following criteria.
> - **Toxicity**. Evaluating toxicity in video generation models is crucial for ensuring user safety, compliance with community standards, and protecting brand reputation.
> - **Fairness.** Assessing fairness in video content ensures a diverse and fair representation of characters and subjects across different social dimensions, preventing the perpetuation of stereotypes and marginalization.
> - **Bias.** Evaluating bias in video generation models is essential for ensuring content accuracy, diversity of perspectives, and avoiding harm to specific groups.
>
> **Conclusion:**
>
> - After studying numerous relevant literature, we use three reference angles, i.e., Toxicity, Fairness, and Bias, to define and measure the subjective metric of "Ethical Robustness".
>
>
> **Q5: How to ensure its applicability across different types of T2V models?**
>
> Thank you for your insightful comment. Regarding the preprocessing problem of different types of videos, we further analyze this from the following perspectives.
>
> **Analysis:**
>
> - **Discussion on the preprocessing of different types of video.** We reviewed over 200 video generation literature from a variety of fields. It is certain that the vast majority of articles, at least in the field of T2V, **do not further preprocess** the video to be evaluated [3,12,13], as further processing would have an impact on some of the characteristics of the original video, while only simple scaling would instead better preserve and evaluate the **model's original capabilities**. This point can be further illustrated with a few examples below:
>
>   - For models that aim to produce high frame rate video [9], it is **not fair** to sample a uniform frame rate during the evaluation process, as this approach undermines the strength of the model.
>   - For studies specializing in the generation of long videos [10], it is also **unfair** to extract frames or cut short pieces of the generated results to compare with short videos, because there is no guarantee that the displayed results can reflect the original advantages of the study.
>   - For articles emphasizing text alignment [11], if the original video is cropped or watermarks are added, the original visible content may be cropped or blocked, which is also an **unfair** comparison.
>
> - **Video comparison of different lengths.** We do not apply any extra processing to the length of the video, and annotators are asked to view the video in its entirety before annotating it, thus ensuring a fair and reliable comparison. We also experimented with comparing VideoCrafter [14] with Sora [15], which proved that even with video length editing, frame rate processing, etc., the annotators all agreed that Sora was significantly better than VideoCrafter on all dimensions.
>
> **Conclusion:**
>
> - In order not to bias any model in comparison, we use the simplest scaling process, which only guarantees that **all videos can be displayed clearly**. Large-scale literature research confirms the fairness of this approach, because any further processing may **undermine** the advantages of some models when the evaluation scope is very wide.

---

> ### Author Response · Authors · 2024-08-07
> **Rebuttal (4/4)**
>
> **Q6: Would the survey of non-T2V evaluation approaches be beneficial when merging with this technology? With the lack of maturity and rapidly evolving field, would not benefit from a cross-domain understanding from more mature fields?**
>
> Thank you for your insightful comment. We analyze it from the following perspectives.
>
> **Analysis:**
>
> - The guidelines and training proposed in this study can **be easily extended** to the T2I and NLG community studies with only user interface modifications, and similarly, by modifying the automated metrics in the dynamic module, one can also achieve efficient human evaluation of other types of protocols.
> - This study drew on some well-established T2I and NLG human assessment studies [2,16] to construct a more comprehensive protocol in terms of some important principles, such as ensuring clarity of definitions and reliability of annotated results, as well as the design of the subjective indicators, which are lacking in existing T2V assessment studies. However, the design of specific indicators, such as the four objective indicators, was derived from specific T2V research needs.
> - We also plan to design prompt datasets specifically for evaluating T2V models based on the principles of dataset construction in T2I and NLG fields in the future.
>
> **Conclusion:**
>
> - Our protocol builds on previous non-T2V evaluation studies and T2V model studies, and the proposed annotator training principles and dynamic evaluation components can also contribute to the development of non-T2V evaluation studies.
>
> **References:**
>
> [1] All That's 'Human' Is Not Gold: Evaluating Human Evaluation of Generated Text. IJCNLP 2021.
>
> [2] Evaluation of Text Generation: A Survey.
>
> [3] Towards A Better Metric for Text-to-Video Generation.
>
> [4] Ties in paired-comparison experiments: A generalization of the Bradley-Terry model. _Journal of the American Statistical Association_, _62_(317), 194-204.
>
> [5] Rank analysis of incomplete block designs: I. The method of paired comparisons. _Biometrika_, _39_(3/4), 324-345.
>
> [6] Holistic Evaluation of Text-to-Image Models. NIPS 2023.
>
> [7] How well can Text-to-Image Generative Models understand Ethical Natural Language Interventions? EMNLP 2022.
>
> [8] Adopting and expanding ethical principles for generative artificial intelligence from military to healthcare. NPJ Digital Medicine 6, Article number: 225 (2023).
>
> [9] ZeroSmooth: Training-free Diffuser Adaptation for High Frame Rate Video Generation.
>
> [10] ControlVideo: Training-free Controllable Text-to-Video Generation. ICLR 2024.
>
> [11] Seer: Language Instructed Video Prediction with Latent Diffusion Models. ICLR 2024.
>
> [12] VBench: Comprehensive Benchmark Suite for Video Generative Models. CVPR 2024.
>
> [13] AIGCBench: Comprehensive Evaluation of Image-to-Video Content Generated by AI.
>
> [14] VideoCrafter2: Overcoming Data Limitations for High-Quality Video Diffusion Models. TBench 2024.
>
> [15] OpenAI Sora.
>
> [16] Toward Verifiable and Reproducible Human Evaluation for Text-to-Image Generation. CVPR 2023.

---

> > ### Comment · Reviewer_AEKZ · 2024-08-08
> >
> > To keep the authors updated:
> >
> > I confirm the reception of the rebuttals. I will follow up soon with any queries and/or updated scores to reflect my understanding.

---

> > > ### Author Response · Authors · 2024-08-10
> > > **Looking forward to further discussion**
> > >
> > > Thank you for confirming the reception of our rebuttals. We appreciate your attention to our submission.
> > >
> > > Please let us know if you have any further queries or require additional information. Your guidance is invaluable to us, and we are eager to address any remaining concerns.
> > >
> > > Best regards

---

> ### Author Response · Authors · 2024-08-11
> **To save reviewer's time, we put a summary of rebuttal**
>
> Dear Reviewer AEKZ,
>
> Thanks so much again for the time and effort in our work. Considering the limited time available and to save the reviewer's time, we **summarized our responses here**.
>
> 1. \[**Lacks an understanding of evaluation in other domains** \]:
>
>    **Response:**
>    - Our protocol is based on a thorough investigation of human evaluation and video generation articles in different fields and is experimentally validated to have the advantages of reproducibility, reliability, and practicability compared to traditional protocols.
>
> 2. \[**Dynamic evaluation module may introduce bias** \]:
>
>    **Response:**
>    - Theoretically, our protocol design allows for plausible estimation in the absence of numerous comparisons, and we further validate this capability through bootstrap confidence interval experiments..
>
> 3. \[**Lack detailed analysis of cases where the protocol fails. Need further evaluation and consideration in other domains** \]:
>
>    **Response:**
>    - We analyzed the case of producing inconsistent results under the traditional protocol setup, demonstrating the importance of the annotator training we designed, and conducted a reliability analysis of the dynamic evaluation component, which proved that the introduction of this component does not result in inconsistent protocol results.
>    - Our design is not only to provide a more reliable, reproducible, and practical T2V assessment protocol but also to provide ideas and reference materials for human evaluation research in other fields..
>
> 4. \[**Small typo errors** \]:
>
>    **Response:**
>    - Thanks for the reminder. These typo errors are caused by automatic line breaks during latex compilation, not actual errors in the text editor.
>
> 5. \[**How the dynamic evaluation module handles potential biases** \]:
>
>    **Response:**
>    - From a methodological design perspective, the Rao and Kupper model we used can accurately estimate model scores using incomplete comparisons.
>    - From an experimental validation perspective, our rigorous bootstrap confidence interval experiments confirm that even after discarding many comparisons on top models such as Gen2, the final estimate of the relative strength of the model is still accurate and credible.
>
> 6. \[**How to capture a representative sample** \]:
>
>    **Response:**
>    - The static evaluation in the protocol retains the samples that are most worthy of being annotated, ensuring that the evaluation results won't be biased by the lack of specific kinds of prompts and model combinations.
>
>
> 7. \[**How to handle potential biases and potential lack of diversity** \]:
>
>    **Response:**
>    - We minimize annotator bias in assessing objective metrics by providing detailed guidelines and training processes, and we retain annotator diversity by introducing subjective metrics..
>
> 8. \[**Provide more details on "Ethical Robustness"** \]:
>
>    **Response:**
>    - Our survey shows that while the measurement of the ethics of generated content is missing from existing T2V evaluation protocols. Therefore, after a thorough investigation of the relevant research, we use three reference angles, i.e., Toxicity, Fairness, and Bias, to define and measure the subjective metric of "Ethical Robustness".
>
> 9. \[**How to ensure its applicability across different types of T2V models** \]:
>
>    **Response:**
>    - In order not to bias any model in comparison, we use the simplest scaling process, which only guarantees that all videos can be displayed clearly.
>    - Large-scale literature research confirms the fairness of this approach, because any further processing may undermine the advantages of some models when the evaluation scope is very wide.
>
> 10. \[**Cross-domain understanding** \]:
>
>    **Response:**
>    - Our protocol builds on previous non-T2V evaluation studies and T2V model studies, and the proposed annotator training principles and dynamic evaluation components can also contribute to the development of non-T2V evaluation studies.
>
>
> Since the discussion stage is already halfway through, may I know if our rebuttal addresses the concerns? If there are further concerns or questions, we are more than happy to address them. Thanks again for taking the time to review our work and provide insightful comments.
>
> Best Regards,
>
> Authors

---

> ### Author Response · Authors · 2024-08-12
> **Looking forward to the reply**
>
> Dear Reviewer AEKZ,
>
> As we draw closer to the rebuttal deadline, we would like to inquire if there are any additional questions or concerns about our work. We greatly value the feedback.
>
> Thanks again for the time and effort in our work!
>
> Best wishes,
>
> Authors

---

> > ### Comment · Reviewer_AEKZ · 2024-08-12
> >
> > I have updated the score, based on the rebuttal presented by the author.

---

> > > ### Author Response · Authors · 2024-08-12
> > > **We appreciate the support!**
> > >
> > > Thanks very much for **supporting and accepting** our work. We are happy that we have addressed your concern. Your constructive feedback is very helpful for us in improving our work.
> > >
> > > If there are any further questions, we are more than willing to provide a detailed explanation. Thank you!

---

### Official Review · Reviewer_2pJH · 2024-07-12

**Soundness:** 3
**Presentation:** 4
**Contribution:** 2
**Rating:** 3
**Confidence:** 5

**Summary:**

This paper designs an evaluation protocol and surveys a number of video generation papers published over the last few years.  The authors point out shortcomings of evaluation protocols in these priors works (e.g. they often limit studies to video quality comparisons with no clear definitions of video quality, small number of videos, annotators etc).  The authors promise to publish their trainings for annotators along with an interface.  They show that these trainings help significantly to improve reliability of annotator scores.

Another contribution is a method that doesn’t require all pairs of models to be compared and dynamically is able to select pairs to be compared and show experimentally that this scales better than the naive quadratic approach.

Finally the authors use their methodology to compare 5 recent models and show among other things that Runway and Pika outperform their open source competitors.

**Strengths:**

Overall the list of surveyed papers in this paper is quite impressive and comprehensive. Moreover, having an evaluation method that can be used without all-pairs of comparisons to be run is clearly important particularly as the number of video generation models increases.  I also absolutely agree with the weaknesses of other video evaluations and agree that human evals are the golden standard and so making a public one available would be very beneficial for the community.

**Weaknesses:**

Weaknesses include the following:
* The authors cite works like Evalcrafter/Vbench but are vague about what value this current submission brings over these prior approaches. Specifically they claim that these approaches “may lack diversity to cover all real-world scenarios” but there are no details given.  As I’ve started to see a number of papers published using the Evalcrafter methodology, I would like to see a clear explanation of the pros and cons.
* Moreover the number of models actually evaluated in this paper is fairly small (only 5) — given that this is an evaluation methodology paper, it would be much stronger to include more models and perhaps even to show clearly that these human evaluations do not correlate well with things like FID or FVD.
* I recommend citing and considering using ELO scores which similarly allow for us to score and rank models based on not-all-pairs comparisons and also allow for dynamic selection of model pairs to be compared side by side.  See e.g., https://lmsys.org/blog/2023-05-03-arena/ for a recent example using ELO to rank LLMs by quality as perceived by humans.  Related is the Microsoft TrueSkill work which has also been used widely.
* Though the authors make a big deal of the fact that few prior works reveal the details of their interface and how the videos are displayed, I found very few details actually revealed in this paper.  The one clear detail I can see is that the authors mention standardizing the height of the video across models.  But this may not be the right thing to do!  Consider the VideoPoet work which generate videos in portrait aspect ratios — resizing so that heights are fixed would be unfair to these vertical videos.  Other details that are important are how to best show videos that are generated at different framerates and even different lengths (for example, how does one compare a 5 second clip from e.g. VideoCrafter to a 1 minute clip from Sora?).
* Finally some things that are marked as objective clearly are not objective like aesthetic quality.  It would be helpful / more convincing to explain clearly what the training looks like for such a dimension.  I would presume that one example showing what aesthetic quality means is not nearly enough…


Minor quibbles:
* I recommend being clear that “higher is better” in many plots
* The authors also use a lot of acronyms which impedes readability in various parts of the paper (e.g. last subsection of 4.3)
* I somewhat disagree with the characterization of FVD being about temporal consistency… even though this is part of it, the statements about the other metrics also apply — it compares feature representations from real and generated images and assesses diversity and clarity…
* I recommend seeing/citing the Videopoet paper that had very similar comparisons.

**Questions:**

See weaknesses above.

**Limitations:**

yes

---

> ### Author Rebuttal · Authors · 2024-08-07
>
> We sincerely thank reviewer 2pJH for the valuable comments and give detailed responses to all questions. Due to space constraints, we submit part of the replies in "Official Comment", for convenience, we have summarized conclusions for each reply. Looking forward to further discussion with you.
>
> **W1: Vague about what value this current submission brings over these prior approaches.**
>
> Thanks for the comments. As we argue in lines 115-124 of the article, Evalcrafter [1], Vbench [2], and other papers have done excellent work in building scorers that match human intuition, as they can implement model evaluation at much lower cost, undoubtedly, these evaluation tools represent a promising research direction, but it cannot completely replace human evaluation protocols for now. We analyze it from the following perspectives.
>
> **Analysis:**
>
> - **Less flexibility and Higher expansion costs.** Since scorers are essentially pre-trained models, though the scorer can reduce evaluation costs, the metrics that can be evaluated and the types of data that can be accepted are fixed. Furthermore, its expansion cost is much higher than our protocol.
> - For example, Vbench's evaluation system includes several pre-trained models to evaluate certain features in the video [2]. If the definition of the existing evaluation object needs to be updated, the evaluation process requires multimodal data [3] or a new evaluation dimension needs to be added, one not only needs to update the project but also needs to conduct large-scale human evaluation experiments to verify its reliability. Meanwhile, updating our human evaluation protocol is **relatively simple**, requiring only an update of the annotation interface.
> - **Scorer training relies heavily on the design and result of human evaluation**. An important measure for all automatic scoring tools is whether they agree with human intuition. Therefore constructing comprehensive and fair human assessment protocols is **an important prerequisite** for training reliable automatic scorers. Our survey shows that many studies training automated scorers use untrained laboratory recruiters, which seriously diminishes the reliability of the scoring results produced by the final trained model. Hence, designing reliable, reproducible, and practical human evaluation protocols could **help improve the quality of such work** as well.
>
> **Conclusions:**
>
> - Although automated scorers are a promising research direction, their inherent poor flexibility, and  high expansion costs result in the fact that they are still **unable to fully replace** human evaluation protocols.
> - More importantly, the design of better human evaluation protocols is itself **an important prerequisite** for the training of more reliable scorers.
>
> **W2: Include more models or show clearly that these human evaluations do not correlate well with things like FID or FVD.**
>
> Thanks for the comments. We analyze it from the following perspectives.
>
> **Analysis:**
>
> - **The number of evaluated models.** We have fully investigated recent benchmark works. Due to the slow inference speed of generative models, all related works focus on evaluating the state-of-the-art four or five models, as shown in the table below, .
>
> | **Method**|T2VScore [4]|EvalCrafter [1]|Vbench [2]|AIGCBench [5]|FETV [6]|T2IHE [7]|
> | -|-|-|-|-|-|-|
> | **Number of evaluated models** | 4| 5| 4| 5| 4| 4|
>
> - **Future work**. Since our work began at the beginning of this year, even though we had selected the most advanced models at the time, as many video generation works were completed in April this year, we still miss many excellent concurrent works. Given the time cost of human evaluation, we are sorry that we cannot add the evaluation results of more models for now. **We will continue to evaluate more models in the future.**
> - **Correlation with FID and FVD.** We utilize the Prompt Suite per Category [2] as a source of prompts, which provides a comprehensive set of prompt types, but as it was built manually, there are **no corresponding reference videos** for calculating FID and FVD. Also, there has been a lot of work demonstrating the inconsistencies between automatic indicators such as FID and FVD and human perception [1,6]. We will use broader prompt sources in the future and further supplement the relevant work.
>
> **Conclusions:**
>
> - We tried our best to evaluate the **five most advanced** video generation works available at the start of the research. This amount is in line with other benchmarking works, and we will conduct more model evaluations later.
> - Since there are **no corresponding reference videos** for the prompt source, we cannot calculate the corresponding FID and FVD indicators, but we will supplement this when expanding our prompt source in the future.

---

> ### Author Response · Authors · 2024-08-07
> **Rebuttal (2/4)**
>
> **W3: Citing and considering using ELO scores.**
>
> Thank you for your insightful comment. We recognize the significance of the ELO rating system and its broad applications in model comparisons. Below, we detail our response and the reasons for choosing the Rao and Kupper model.
>
> **Analysis:**
>
> - **ELO Rating as a Bradley-Terry Special Case:** The ELO rating system can be viewed as a special case of the Bradley-Terry model [8]. It estimates object scores based on the outcomes of pairwise comparisons, and the resulting calculated scores are fed into the sampling algorithm to determine subsequent competition model pairs [10].
> - **Ties and Dynamic Selection:** While the ELO rating system provides a robust framework for scoring and ranking, it doesn't inherently account for ties, which can occur frequently in our dataset. In contrast, the Rao and Kupper model [9], which we adopted, is a modification of the Bradley-Terry model that accommodates tied comparison results. This capability is crucial for our specific use case, where annotators may not always have a clear preference.
> - **Rao and Kupper Model Advantages.**  (1) **Handling Ties:** The Rao and Kupper model's ability to handle tied outcomes allows for more precise scoring and ranking when annotators express equal preference between models. (2) **Dynamic Pair Selection:** Although both models support dynamic pair selection, the Rao and Kupper model's handling of ties makes it more suitable for scenarios where comparisons may not always result in a decisive winner.
> - **Consideration for Future Work:**  We acknowledge the benefits of the ELO system's dynamic selection algorithm and will consider incorporating these insights into our future research to enhance our model ranking process. We will also add discussions and references to similar content in our article revisions.
>
> **Conclusions:**
>
> - While the ELO rating system is powerful, the Rao and Kupper model **better fits the requirements of our task** due to its flexibility in handling ties.
> - The main purpose of our work is to **design reliable, reproducible, and practical human evaluation protocol for T2V models**, but we will also build and maintain a public, large-scale, long-term project in the future.

---

> ### Author Response · Authors · 2024-08-07
> **Rebuttal (3/4)**
>
> **W4: Few details are revealed in this paper. Standardizing the height of the video across models may not be the right thing to do. How to best show videos that are generated at different framerates and even different lengths.**
>
> Thank you for your insightful comment. We provide full details of the protocol in the article and **Appendix**. For your ease of reference, we provide **the location of all details** in the table below.
>
> || **Evaluation metrics**|**Scoring methods**|**Evaluator requirements**|**Dynamic component**|
> |-|-|-|-|-|
> | **Position** | Section 4.1, Line 177-197$ $     Appendix C.4, Line 829, Table 4 | Section 4.2, Line 199-219$ $     Appendix C.3, Line 787-820 | Section 4.3, Line 221-253 | Section 4.4, Line 254-271$ $      Appendix C.2, Line 737-786 |
>
> || **Annotator training** |**Annotation interface** | **Details of the models evaluated**| **Details of the videos evaluated** |
> |-|-|-|-|-|
> | **Position** | Section 4.3, Line 230-234 $ $     Figure 1 (c) and Figures 5 to 9 | Section 1, Line 41-45$ $     Figure 1 (b) | Section 5, Line 273-276$ $    Appendix C.1, Line 727-736 | Section 5.1, Line 278-283 |
>
> We've revised the article to include more detailed information in the body of the article and will **open source** all the relevant interfaces and code. In addition, we are willing to provide **any information** related to the protocol to help the community.
>
> Regarding the preprocessing problem of different types of videos, we further analyze this from the following perspectives.
>
> **Analysis:**
>
> - First of all, even if VideoPoet [19] produces videos in portrait aspect ratios, its characteristics such as clarity in the evaluation interface **won't be affected**.
> - Specifically, T2VHE take the highest video among the videos to be evaluated as the video height displayed in the annotation interface, and zoom out of the videos smaller than this height, so as to guarantee that **every video can be displayed clearly**. We will add an explanation of this issue in Section 5.1 to avoid this misunderstanding.
> - **Discussion on the preprocessing of different types of video.** We reviewed over 200 video generation literature from a variety of fields. The vast majority of articles, at least in the field of T2V, **do not further preprocess** the video to be evaluated [2,4,5], as further processing would have an impact on some of the characteristics of the original video, while only simple scaling would instead better preserve and evaluate the **model's original capabilities**. This point can be further illustrated with a few examples below:
>
>   - For models that aim to produce high frame rate video [11], it is **not fair** to sample a uniform frame rate during the evaluation process, as this approach undermines the strength of the model.
>   - For studies specializing in the generation of long videos [12], it is also **unfair** to extract frames or cut short pieces of the generated results to compare with short videos, because there is no guarantee that the displayed results can reflect the original advantages of the study.
>   - For articles emphasizing text alignment [13], if the original video is cropped or watermarks are added, the original visible content may be cropped or blocked, which is also an **unfair** comparison.
>     It must be admitted that the preprocessing methods used in the various articles may be fairer for  particular comparisons, but to fit the protocol into the extensive T2V studies, simply scaling the video is a viable approach for human evaluation protocols.
> - **Video comparison of different lengths.** We do not apply any extra processing to the length of the video, and annotators are asked to view the video in its entirety before annotating it, thus ensuring a fair and reliable comparison. We also experimented with comparing VideoCrafter [14] with Sora [15], which proved that even with video length editing, frame rate processing, etc., the annotators all agreed that Sora was significantly better than VideoCrafter on all dimensions.
>
> **Conclusion:**
>
> - In order not to bias any model in comparison, we use the simplest scaling process, which only guarantees that **all videos can be displayed clearly**. Large-scale literature research confirms the fairness of this approach, because any further processing may **undermine** the advantages of some models when the evaluation scope is very wide.

---

> ### Author Response · Authors · 2024-08-07
> **Rebuttal (4/4)**
>
> **W5: Explain clearly what the training looks like for aesthetic quality.**
>
> Thank you for your insightful comment. We analyze it from the following perspectives.
>
> **Analysis:**
>
> - We provide a **detailed display** of the training content in Figure 1 (c) and Figures 5 to 9. Specifically, for each reference angle, we provide a set of **reference videos** with the **corresponding analysis process**. For your reference, a detailed description of the "Aesthetic Appeal" and corresponding training examples can be found in Table 4 and Figure 1 (c).
> - **Objective and subjective metrics are relative.** Our division between subjective and objective metrics stems mainly from the fact that for subjective metrics, such as human preferences, we cannot find a universally recognized and workable description for annotators to adhere to. Whereas for objective metrics, such as "Aesthetic Appeal", several studies have rigorously defined these [2,18], we can design effective training examples based on previous research and ask the annotators to try to be objective in their judgments.
> - **Annotators are different from automatic scorers**. While they can understand the concept of more abstract indicators, **no amount of detailed guidance is enough** to get them rid of their subjective influence [7,16,17]. However, this effect can be reduced through annotator training. As shown in Table 1, there is better consistency in the annotators after the training.
>
> **Conclusions:**
>
> - The distinction between subjective and objective indicators stems from whether we can ask the annotator to evaluate according to predefined, accepted, and feasible criteria.
> - Although it is impossible to achieve a completely objective human assessment, experiments show that we can **effectively reduce the influence** of personal preferences on results through well-designed assessment protocols and detailed annotator training.
>
>
> **M1: Being clear that “higher is better” in many plots.**
>
> Thank you for your comment, we have added that in the table captions.
>
> **M2: Using a lot of acronyms may impede readability in various parts of the paper.**
>
> Thanks to your suggestion, we have reduced using acronyms in the text, particularly in section 4.3.
>
> **M3: Disagree with the characterization of FVD being about temporal consistency.**
>
> Thank you for your comments, we have amended the description of the FVD indicator within the related work section.
>
> **M4:Recommend seeing/citing the Videopoet paper.**
>
> As the survey covers video generation work in many areas such as T2V, video editing, frame prediction, and motion control, as well as human evaluation work in related areas such as T2I, NLG, and so on, we apologize for neglecting such excellent work as Videopoet [19] in our research. Thanks for the reminder, we've read the work carefully and cited it in the revised article.
>
> **References:**
>
> [1] EvalCrafter: Benchmarking and Evaluating Large Video Generation Models. CVPR 2024.
>
> [2] VBench: Comprehensive Benchmark Suite for Video Generative Models. CVPR 2024.
>
> [3] Show Me What and Tell Me How: Video Synthesis via Multimodal Conditioning. CVPR 2022.
>
> [4] Towards A Better Metric for Text-to-Video Generation.
>
> [5] AIGCBench: Comprehensive Evaluation of Image-to-Video Content Generated by AI. TBench 2024.
>
> [6] FETV: A Benchmark for Fine-Grained Evaluation of Open-Domain Text-to-Video Generation. NIPS 2023.
>
> [7] Toward Verifiable and Reproducible Human Evaluation for Text-to-Image Generation. CVPR 2023.
>
> [8] Rank analysis of incomplete block designs: I. The method of paired comparisons. _Biometrika_, _39_(3/4), 324-345.
>
> [9] Ties in paired-comparison experiments: A generalization of the Bradley-Terry model. _Journal of the American Statistical Association_, _62_(317), 194-204.
> [10] Chatbot Arena.
>
> [11] ZeroSmooth: Training-free Diffuser Adaptation for High Frame Rate Video Generation.
>
> [12] ControlVideo: Training-free Controllable Text-to-Video Generation. ICLR 2024.
>
> [13] Seer: Language Instructed Video Prediction with Latent Diffusion Models. ICLR 2024.
>
> [14] VideoCrafter2: Overcoming Data Limitations for High-Quality Video Diffusion Models. TBench 2024.
>
> [15] OpenAI Sora.
>
> [16] All That's 'Human' Is Not Gold: Evaluating Human Evaluation of Generated Text. IJCNLP 2021.
>
> [17] Evaluation of Text Generation: A Survey.
>
> [18] Holistic Evaluation of Text-to-Image Models. NIPS 2023.
>
> [19] VideoPoet: A Large Language Model for Zero-Shot Video Generation. ICML 2024.

---

> ### Author Response · Authors · 2024-08-11
> **To save reviewer's time, we put a summary of rebuttal**
>
> Dear Reviewer 2pJH,
>
> Thanks so much again for the time and effort in our work. Considering the limited time available and to save the reviewer's time, we **summarized our responses here**.
>
> 1. \[**Value over prior approaches** \]:
>
>    **Response:**
>    - While automated scorers can implement evaluations at lower cost, their inherent poor flexibility, and high expansion costs result in the fact that they are still unable to fully replace human evaluation protocols.
>    - The design of better human evaluation protocols is itself an important prerequisite for the training of more reliable automated scorers.
>
> 2. \[**Include more models or show human evaluations do not correlate FID or FVD** \]:
>
>    **Response:**
>    - We evaluate the five most advanced video generation works available at the start of the research. This amount is in line with other benchmarking works, and we will conduct more model evaluations later.
>    - Since there are no corresponding reference videos for the prompt source, we cannot calculate the corresponding FID and FVD indicators, but we will supplement this when expanding our prompt source in the future.
>
> 3. \[**Citing and considering using ELO scores** \]:
>
>    **Response:**
>    - While the ELO rating system is powerful, our Rao and Kupper model better fits the requirements of our task due to its flexibility in handling ties.
>    - We will also build and maintain a public, large-scale, long-term project in the future.
>
> 4. \[**Few details. How to best show videos with different characteristics** \]:
>
>    **Response:**
>    - In our response Rebuttal (3/4), we provided the locations of all the protocol details revealed and we've revised the article to include more detailed information in the body of the article.
>    - We use the simplest scaling process to avoid biasing any model in comparison and guarantee that all videos can be displayed clearly.
>    - Large-scale literature research confirms the fairness of this approach, because any further processing may undermine the advantages of some models when the evaluation scope is very wide.
>
> 5. \[**Some things marked as objective are not objective** \]:
>
>    **Response:**
>    - The distinction between subjective and objective indicators stems from whether we can ask the annotator to evaluate according to predefined, accepted, and feasible criteria.
>    - Experiments show that we can effectively reduce the influence of personal preferences on results through well-designed assessment protocols and detailed annotator training.
>
> 6. \[**Minor quibbles** \]:
>
>    **Response:**
>    - Thanks to your suggestion, we have revised the article to address the concerns raised.
>
>
> Since the discussion stage is already halfway through, may I know if our rebuttal addresses the concerns? If there are further concerns or questions, we are more than happy to address them. Thanks again for taking the time to review our work and provide insightful comments.
>
> Best Regards,
>
> Authors

---

> ### Author Response · Authors · 2024-08-12
> **Looking forward to the reply**
>
> Dear Reviewer 2pJH,
>
> As we draw closer to the rebuttal deadline, we would like to inquire if there are any additional questions or concerns about our work. We greatly value the feedback.
>
> Thanks again for the time and effort in our work!
>
> Best wishes,
>
> Authors

---

> ### Author Response · Authors · 2024-08-13
> **Looking forward to the reply**
>
> Dear Reviewer 2pJH,
>
> We appreciate your constructive comments and have **responded in detail to the five weaknesses**, while we also **revised the article** to address the minor quibbles raised.
>
> As we draw closer to the rebuttal deadline, we would like to inquire if there are any additional questions or concerns about our work. We greatly value the feedback.
>
> Thanks again for the time and effort in our work!
>
> Best wishes,
>
> Authors

---

### Decision · Program_Chairs · 2024-09-25

**Decision:**

Accept (poster)

**Comment:**

This paper received a Strong Accept, 2 Borderline Accept and a Reject. The reviewers found the paper novel, well motivated and the references to prior work quite thorough and beneficial to the community. The main concerns were that: 1) it is unclear what additional benefit the proposed method brings over prior work, 2) the limited comparisons, 3) the design of standardisation is not satisfactory (ie, how to compare videos with different aspect ratios, length, fps), 4) lack of error analysis. The authors engaged in extensive discussions with the reviewers. 3 reviewers acknowledged that the authors' responses were satisfactory. 2 raised their ratings, while one kept his high score.
The reviewer who recommended a Reject for the paper did not engage in the discussion nor provided further input to clarify if the answers addressed the concerns. The AC has read all the reviews and the discussions and finds that the authors addressed all the concerns. Thus the AC recommends this paper for acceptance.